

# Contribution of low-frequency climatic/oceanic oscillations to streamflow variability in small, coastal rivers of the Sierra Nevada de Santa Marta (Colombia)

Juan Camilo Restrepo [1], Aldemar Higgins [1], Jaime Escobar [2,3], Silvio Ospino [1], Natalia Hoyos [4]

[1]Grupo de Investigación en Geociencias GEO4, Departamento de Física y Geociencias, Universidad del Norte, km 5 vía Puerto Colombia, Barranquilla - Colombia.
[2]Departamento de Ingeniería Civil y Ambiental, Universidad del Norte, km 5 vía Puerto Colombia, Barranquilla - Colombia.
[3]Smithsonian Tropical Research Institute, Box 0843-03092, Balboa, Ancon, Republic of Panama.
[4]Departamento de Historia y Ciencias Sociales, Universidad del Norte, km 5 vía Puerto Colombia, Barranquilla - Colombia.

*Correspondence to*: Juan C. Restrepo (restrepocj@uninorte.edu.co)

**Abstract.** This study evaluated the influence of low-frequency oscillations that are linked to large-scale oceanographic/atmospheric processes, on streamflow variability in small tropical coastal mountain rivers of the Sierra Nevada de Santa Marta, Colombia. We used data from six rivers that had >32 years of complete, continuous monthly streamflow records. This investigation employed spectral analyses to (1) explore temporal characteristics of streamflow variability, (2)

estimate the net contribution to the energy spectrum of low-frequency oscillations to streamflow anomalies, and (3) analyze the linkages between streamflow anomalies and large-scale, low-frequency oceanographic/atmospheric processes. Wavelet analyses indicate that the 8-12-yr component exhibited a quasi-stationary state, with a peak of maximum power between 1985 and 2005. These oscillations were nearly in phase in all rivers. Maximum power peaks occurred for the Palomino and Rancheria Rivers in 1985 and 1995, respectively. The wavelet spectrum highlights a change in river variability patterns between 1995

and 2015, characterized by a shift towards the low-frequency oscillations domain (8-12 yr). The net contribution of these oscillations to the energy spectrum was as high as 51%, a value much larger than previously thought for rivers in northwestern South America. The simultaneous occurrence of hydrologic oscillations, as well as the increase in the amplitude of the 8-12-yr band, defined periods of extremely anomalous wet seasons during 1989-1990, 1998-2002 and 2010-2011, reflecting the role of low-frequency oscillations in modulating streamflow variability in these rivers. Cross Wavelet Transform and Wavelet

Coherence revealed high common powers and significant coherences in low-frequency bands (>96 months) between streamflow anomalies and Atlantic Meridional Oscillation (AMO), Pacific Decadal Oscillation (PDO) and the Tropical North Atlantic Index (TNA). These results show the role of large-scale, low-frequency oceanographic/climate processes in modulating long-term hydrological variability of these rivers.





# 1 Introduction

In the past several decades, streamflow variability has increased (Milliman et al., 2008; Dai et al., 2009), causing frequent and pronounced flood/drought cycles (Hungtinton, 2006). Atmospheric and oceanographic processes are major sources of streamflow variability (Jhonson et al., 2013; Schulte et al., 2016). The El Niño-Southern Oscillation (ENSO) is among the most prevalent oceanographic/atmospheric processes linked to streamflow variability in northern South America (Battisti and Sarachick, 1995; Amarasekera et al., 1997; García and Mechoso, 2005). ENSO, however, is also affected by longer-period changes in the background state (Garreaud et al., 2009; Chowdary et al., 2014). Thus, a major question in the study of hydrology in northern South American is the potential effect of longer-period climate modes, such as the Pacific Decadal Oscillation (PDO), on the strength of a particular El Niño/Niña event. For example, the 1997-1998 El Niño event occurred during a PDO shift from a warm to a cold phase, but recent warming (2010-2011) in the Pacific occurred during a cold phase of the PDO. Such atmospheric and oceanographic interactions, as well as their role in hydrological variability, have gained attention in recent years (Tootle et al., 2008; Arias et al., 2015; Sagarika et al., 2015; Nalley et al., 2016). For example, the extremely anomalous wet seasons in northern South America between 2010 and 2012 were associated not only with ENSO anomalies, but also with an enhanced Atlantic Meridional Mode (AMO), a low-frequency oscillation that is independent of ENSO (Arias et al. 2015).

Spectral analyses such as Wavelet Transform (WT) and the Hilbert Huang Transform (HHT) (Grinsted et al., 2004; Labat et al., 2005; Torrence and Compo, 2008; Massei and Fournier, 2012; Schulte et al., 2016) have proven useful to identify the timing of important features of non-stationary signals and to discriminate the relative contribution of signal components, which may change through time. Contribution from low-frequency oscillations to streamflow variability is poorly understood, particularly in small, tropical, coastal mountain rivers (Stevens and Ruscher, 2014; Nalley et al., 2016; Marini et al., 2016). These fluvial systems possess low streamflow buffering capacity because of their topographic setting (Milliman and Syvitski, 1992), and they are exposed to regional-scale atmospheric/oceanographic processes (Hastenrath, 1990; Enfield and Alfaro, 1999). Several authors have examined the relationship between streamflow variability in northern South America and large-scale oceanographic/climate indices, particularly those linked to ENSO (e.g. the Southern Oscillation Index [SOI], the Multivariate ENSO Index [MEI], and Niño 1, 2, 3, 4) (Robertson and Mechoso, 1998; Hastenrath, S., 1990; Gutiérrez and Dracup, 2001; Poveda et al., 2001; Restrepo and Kjerfve, 2004; García and Mechoso, 2005). New variables such as SST gradients in the Caribbean Sea and low-frequency oscillations, together with new statistical methods (e.g. Singular Value Decomposition and Principal Components Analyses) are now used in streamflow analysis. These new approaches have improved hydrological forecast models, compared to predictions based solely on El Niño-based indices. The new models also reduce the spatial bias of SST, which affects hydrology at regional scales (Tootle et al., 2008; Córdoba-Machado et al., 2016). These studies, however, failed to include representative small basins (area $\leq 5000$ km$^2$) that drain into the Caribbean Sea in northern South America. Furthermore, mountain rivers flowing from the Sierra Nevada de Santa Marta (SNSM) massif (Fig. 1, Table 1) are absent from these models. Pierini et al. (2015) indicated that rivers from the SNSM exhibit a distinctive



hydrological pattern, which differs from that of other rivers in northwestern South America. Differences are especially pronounced between rivers in the SNSM and those with headwaters in the Colombian Andes. The main difference lies in the relatively low contribution from ENSO-related oscillations to the net streamflow variability exhibited by SNSM rivers (Restrepo et al., 2012, 2014). Furthermore, it has been established that changes in the Caribbean SST gradients affect the

amount of rainfall in northern South America (Enfield and Alfaro, 1999), but there is no evidence that such changes affect the hydrological variability of SNSM rivers, which are characterized by a limited ability to filter hydrological signals (Restrepo et al., 2014).

The objectives of this study were to: (1) study the influence of low-frequency oscillations (linked to large-scale oceanographic/atmospheric processes) on streamflow variability, (2) explore the temporal characteristics of streamflow

variability, (3) estimate the net contribution (i.e. energy spectrum) of low-frequency oscillations to streamflow anomalies, and (4) analyze the linkages between streamflow anomalies and large-scale, low-frequency, oceanographic/atmospheric processes (Table 2) in small, tropical, coastal mountain rivers of the SNSM (Fig. 1 and Table 1). To our knowledge, this is the first study to estimate the contribution of low-frequency oscillations to the hydrologic variability in such small, tropical coastal, mountainous watersheds.

## 2 Study Area

The Sierra Nevada de Santa Marta is a massif of metamorphic and intrusive rocks that is isolated from other mountain ranges that make up the Andean Colombian Cordillera (Montes et al., 2010) (Fig. 1). The SNSM possesses the highest peak in Colombia (5800 m) and is contiguous with an ocean trench about 3200 m deep. These two features form one of the greatest

topographic gradients of any coastal range in the world. Rivers with areas of less than 5000 km$^2$, very steep slopes and small alluvial flood plains drain the Sierra with only the exception of the Rancheria River, which drains large lowlands areas in the Guajira Peninsula (Fig. 1). Rivers that run through the western slopes of the SNSM flow into the Ciénaga Grande de Santa Marta (CGSM), the largest Colombian coastal lagoon (~730 km$^2$), which was designated a RAMSAR site because of its ecological importance (Fig. 1). The environmental and ecological functioning of the CGSM depends heavily on water

discharge from the rivers on the western slopes of the SNSM (Vilardy et al., 2011). Rivers that drain the northern and eastern slopes discharge directly into the Caribbean Sea. Their fluvial discharge play an important role in beach stability (Restrepo et al., 2017). Rivers that drain the southern slopes were not included in this study as they are not considered coastal rivers and/or are tributaries of high-order rivers.

The SNSM experiences two wet seasons annually, as a result of meridional displacement of the Intertropical Convergence

Zone (ITCZ). Precipitation from May to June is associated with northward movement of the ITCZ. Rain, of higher intensity and duration, extends from September to November, when the ITCZ moves southward. The dynamics of the North Jet (i.e. Caribbean or San Andrés Jet), and mountain effects produced by the SNSM, produce a more complex regional distribution of

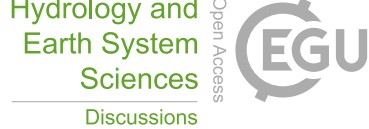



moisture (Bernal et al., 2006; Poveda et al., 2001). The interaction between the northeast trade winds and belts of low-pressure at <900 hPa between latitudes 13°N - 14°N promote formation of the North Jet stream. This jet stream produces a deflection of moisture in northwestern South America, which leads to the rise of air masses along the slopes of the SNSM and causes strong surface wind currents and low humidity on the Guajira Peninsula (Bernal et al., 2006). Thus, the SNSM experiences

precipitation of >2000 mm yr$^{-1}$ and a mean annual temperature of <20 °C. The rest of the Colombian Caribbean is warm and dry, with rainfall <1000 mm yr$^{-1}$ and mean annual temperatures >27 °C (Poveda, 2004).

Mean monthly streamflow in the principal rivers of the SNSM ranges between 2.3 and 28.5 m$^3$ s$^{-1}$ (Fig. 1 and Table 1). These rivers exhibit strong seasonal variability in their discharge, usually as high as 5-10 folds when comparing lowest and highest monthly streamflow. Inter-annual flow variability can also be large, with values for high-streamflow years 2-4x those during

low-streamflow years (Restrepo et al., 2014). In addition, the SNSM rivers exhibit high to very high discharge variability ($Q_{max}/Q_{min}$), high flood regime ($Q_{max}/Q$), while possessing drainage areas < 5.0 x10$^3$ km$^2$ in mountainous zones (Table 1). Thus, topography is a primary factor controlling flood variability (Restrepo et al., 2014).

## 3 Data and Methods

### 3.1 Streamflow Data

Streamflow monthly average data from rivers that drain the SNSM were obtained from the Colombian Hydrology and Metereology Institute (IDEAM). Streamflow data are useful to understand the hydrologic response of a watershed to climate variability (Dai et al., 2009; Labat, 2010) as they integrate stream response to multiple hydrologic processes (precipitation, groundwater exchange, evapotranspiration and runoff) (Milliman et al., 2008). Selection of streamflow gauging stations was

based on the length of records, which had to be sufficiently long to enable analysis of the role and properties of low-frequency oscillations on streamflow variability. For this study, the quasi-decadal oscillations (i.e. 8-12 years) or longer-period oscillations, were classified as low-frequency oscillations. This classification is in close agreement with several previous hydrologic studies (Robertson and Mechoso, 1998; Pekarova et al., 2003; Grinsted et al., 2004). Only stations with a minimum of 32 years of data were selected, to obtain statistically significant information about quasi-decadal oscillations, following the

edge effects ($T/\sqrt{2}$) and the cutoff frequency ($T/2$) approach, where $T$ is the hydrological record total length (Shumway and Stoffer, 2004). We used streamflow data from six rivers with more than 32 years of complete monthly records (Fig. 1 and Table 3).

We used Continuous Wavelet Transform (CWT) and Hilber Huang Transform (HHT) analyses to estimate periodicities, variability patterns, and the net contribution (*i.e.* energy spectrum) of low-frequency oscillations to streamflow anomalies. We

also used Wavelet Coherence (WTC) and Cross Wavelet Transform (XWT) to estimate the correlation between streamflow and eight large-scale climate/oceanographic processes (Table 2) (Shumway and Stoffer, 2004; Grinsted et al., 2004; Labat, 2005). Definitions of the climate/oceanographic indices used in this study are presented in Table 2. Only the most significant



results (PDO, AMO, and TNA) are shown, with other results displayed in Supplementary Material. Prior to the time series analyses continuity and homogeneity tests were applied. Data series with a non-normal distribution were transformed prior to applying the XWT and WTC analyses (Grinsted et al., 2004).

## 3.2 Spectral Wavelet analysis

Analysis of monthly streamflow data was performed using generalized local base functions with Continuos Wavelet Transform (CWT). Mother wavelets were translated and stretched in time resolution and frequency (Torrence and Compo, 1998). This technique is helpful with the evaluation of time series that contain non-stationary functions with different frequencies and provides a time-scale signal localization. The CWT, applied to monthly, "de-seasonalized" streamflows, was used to estimate

periodicities and variability patterns, distinguish temporal oscillations, and identify the intermittency of each time-scale process. The wavelet spectrum was also time averaged (global wavelet spectrum) in order to quantify the main scales of underlying processes, and to determine the signal distribution variance. The global wavelet spectrum provides an adequate estimation of the long-term processes characteristics (Torrence and Compo, 1998; Labat, 2005). The XWT and the WTC were estimated based on the CWT. An XWT spectrum between signals indicates regions where there is common high power and

reveals information about phase relationships. The WTC spectrum highlights the intensity of the covariance of these signals, regardless of the high-power display (Grinsted et al., 2004; Nalley et al., 2016). The WTC covariance ranges from 0 to 1, where 1 represents the highest covariance. Values of the coherence coefficient were estimated following the Grinsted et al. (2004) procedure. The relationship between streamflow and some large-scale oceanographic/atmospheric indices were identified using the phase angle observed in the spectra. An in-phase relationship is indicated by arrows in the enclosed

significant regions of the XWT and WTC spectra that point straight to the right. On the other hand, and anti-phase relation is indicated by arrows pointing straight to the left. Arrows that do not point straight to the right or left indicate a lead/lag relationship, when a climate/oceanographic index led the streamflow response (Grinsted et al., 2004; Nalley et al., 2016).
A complex symmetric function, the Morlet wavelet spectrum, was used to distinguish between the real and imaginary wavelet parts. The imaginary section contains the phase information, that is necessary to calculate the spectrum coherence between

two variables (WTC) (Grinsted et al., 2004; Nalley et al., 2016). The real section captures the negative a positive time series oscillatory characteristics and isolates components such as jumps and discontinuities. The Morlet wavelet also allows to describe the hydrological data structure as well as to have better frequency resolution (Grinsted et al., 2004; Labat, 2005). A value of 6 was defined for the frequency localization of the Morlet wavelet ($\omega_o$) to fulfill the admissibility condition (localization in time and frequency, zero mean, and to acquire a proper balance between frequency and time) (Torrence and

Compo, 1998; Grinsted et al., 2004; Nalley et al., 2016). The 95% confidence level was calculated for contours and edge effects area after the method of Torrence and Compo (1998). The edge effect was addressed by the zero-padding approach. This procedure creates discontinuities at both ends of the data, particularly at larger scales. The power displayed in this area is





expected to be weaker than actually shown (Nalley et al., 2016). The area in the WT spectrum where the edge effect is shown is referred to as the Cone of Influence (COI). The interpretation of the WT power spectra was limited to the area outside the COI, thus the COI is represented by the region outside of the concave-up area.

5 **3.3 Hilbert Huang Transform**

The Hilbert Huang transform (HHT) is an adaptive empirical method used to obtain modes of variability with non-linear and non-stationary data (Huang et al., 1999). The additive decomposition of a time series ($X_{(t)}$) is obtained from the Intrinsic Modes Functions (IMFs) and the residual (Eq.1) indicates the data trend. This process is known as the Empirical Mode Decomposition (EMD). The time series ($X_{(t)}$) can be represented by the sum of the modes ($c_i$) plus the residue ($r_{n(t)}$).

$$X_{(t)} = \sum_{i=1}^{n-1} c_i + r_{n(t)} \tag{1}$$

where $c_i$ represents each of the decomposition modes, $r$ is the residue and $n$ is the number of decomposition modes. The Hilbert transform ($\bar{C}_{i(t)}$) (Eq. 2) is then applied to each of the IMFs to extract the information in terms of energy-time-frequency,

$$\bar{C}_{i(t)} = \frac{1}{\pi} P \int_0^\infty \frac{c_{i(t')}}{t-t'} dt' \tag{2}$$

where $P$ is the main value of the Couchy number (Long et al., 1995). We can then construct the analytic signal ($C_{i(t)}$) from the $\bar{C}_{i(t)}$, defined as (Eq. 3),

$$C_{i(t)} = c_{i(t)} + j\bar{C}_{i(t)} = A_{(t)} e^{-j\theta_{i(t)}} \tag{3}$$

where $A_{(t)} = \left[ c_{i(t)}^2 + \bar{C}_{i(t)}^2 \right]^{1/2}$ and $\theta_{i(t)} = Arctan\left[\frac{\bar{C}_{i(t)}}{c_{i(t)}}\right]$, which corresponds to the instantaneous amplitude and the instantaneous phase angle, respectively. The instantaneous frequency ($w_{i(t)}$) associated with each IMF is defined as (Eq. 4),

$$w_{i(t)} = \frac{1}{\pi} \frac{d\theta_{i(t)}}{dt} \tag{4}$$

The original signal (Eq. 5), excluding the residue, can be expressed from the real part, i.e. the left side of Eq. 3,

30 $$X_{(t)} = Re\left[\sum_{i=1}^N Ai_{(t)} e^{-j\theta_{i(t)}}\right] = Re\left[\sum_{i=1}^N Ai_{(t)} e^{-j\int wi_{(t)} dt}\right] \tag{5}$$





The Hilbert spectrum can be expressed from the square of the instantaneous amplitude ($H(w,t) = A^2(w,t)$). From the Hilbert spectrum we can define the Hilbert marginal spectrum ($h(w)$) (Eq. 6), which represents the sum of all amplitudes (energy) over all data (Barnhart, 2011),

$$h(w) = \int_0^T H(w,t)dt \qquad (6)$$

The Hilbert marginal spectrum corresponds to the energy associated with each of the frequencies that make up the signal (Huang et al., 1999). In order to determine the frequency and energy of each of the signal modes we define the average frequency ($f(n)$) and average energy ($E_n$) of each mode (Huang et al., 1999, 2009),

$$\bar{f}(n) = \frac{\int_0^\infty fE_n(f)df}{\int_0^\infty fE_n(f)df} \qquad (7)$$

$$E_n = \int_0^T C_{n(t)}^2 dt \qquad (8)$$

where $E_n$ is the Fourier power density spectrum. A combination of the algorithm to obtain the functions of intrinsic modes, together with the Hilbert spectral analysis, is called the Hilbert-Huang transform (Huang et al., 1999; Barnhart, 2011).

## 4 Results

### 4.1 Short- and Long-Term Patterns of Streamflow Variability

Intra-annual and annual processes are the dominant signals, but they exhibit intermittency throughout the CWTs (Fig. 2). The Fundación, Aracataca, Palomino and Ranchería Rivers exhibit an intermittent signal at the intra-annual and annual bands, with maximum powers localized around the periods 1985-1990, 1998-2002 and 2008-2010. The Frío and Gaira Rivers show a quasi-continuous annual signal of comparable magnitude, with maximum powers around the 1985-2010 and 1985-2002 intervals, respectively (Fig. 2). In most of these rivers, the inter-annual signal (i.e. 3-7 years) was discontinuous, highly localized, and exhibited relatively low powers throughout the CWT spectra (Fig. 2). These spectra highlight inter-annual processes in the Fundación, Aracataca and Gaira Rivers during the 1995-2005 period, whereas the Ranchería River experienced such process over the 1980-1988 and 1995-2005 intervals. The inter-annual signal also appeared in the Frío and Palomino Rivers from 1996/1998 to 2010. Most inter-annual signals exhibit their maximum power around the 1998- 2002 interval (Fig. 2). Quasi-decadal or low-frequency signals (i.e. >8-year band) are observed in most CWT spectra. All rivers exhibit powers



of comparable magnitude between the 1980s and 2010. The intensity of this signal increased in the 1990s and reached maximum power during the 1998-2005 interval (Fig. 2).

Most of the CWT spectra exhibit periods where the maximum power of the different signal bands occurs simultaneously (Fig. 2). For example, all rivers exhibit high power signals on the annual and quasi-decadal bands during the 1988-1990 interval.

All rivers, except the Palomino, experienced superimposed oscillations on the annual, inter-annual and quasi-decadal bands over the 1998-2002 interval. A quasi-biennial oscillation occurred jointly with annual, inter-annual and quasi-decadal oscillations during the 2008-2012 interval, in the Fundación, Aracataca, Frío and Palomino Rivers (Fig. 2). The simultaneous occurrence of relatively high-power signals led to periods of intense hydrological variability, in which extreme flows occurred, such as those experienced in most rivers in 1988-1989, 1998-2000 and 2010-2011 (Fig. 1 and 2).

The global wavelet spectrum is obtained by time-averaging processes of the CWT (Fig. 3). In the Frío, Gaira and Ranchería Rivers, the main oscillatory component is the annual band. The second-order source of hydrologic variability is the quasi-decadal band. The global wavelet spectrum also displays an 8-12-year or larger oscillation, which constitutes the main oscillatory component in the Fundación, Aracataca and Palomino Rivers. Although the inter-annual scales are common in all rivers, excluding the Aracataca River, they exhibit relatively low power signals and thus constitute a second-order source of

hydrologic variability (Fig. 3). Oscillations greater than one year were not statistically significant, except in the Aracataca River. The global wavelet spectrum remains unchanged (dashed gray lines in Fig. 3) when the intra-annual and annual bands are extracted, removing monthly average values from the original streamflow time series. The significance-level curve, however, changes, making the power of certain oscillations, such as in the Fundación and Frío Rivers, significant (Fig. 3). Thus, it is likely that longer time series are required to test the low-frequency oscillations statistical significance within the

global wavelet spectrum. Information on these low-frequency oscillations was considered useful because (1) the zero-padding technique reduces the lower frequencies true power, (2) the CWT isolates hidden signals not shown by other techniques, and (3) they are within the range defined byedge effects and cut-off frequency.

## 4.2 Intrinsic Component of Monthly Streamflow and Energy Distribution

IMFs were generated by extracting the monthly, multi-year average from the original time series data. We obtained between 6 and 8 oscillation modes and the residual (Fig. 4 and Table 4). The C1-C4 modes represent intra-annual, annual and quasi-biennial oscillations. Mode C5 represents the inter-annual oscillations. Mode C6 and higher modes correspond to low-frequency oscillations (i.e. quasi-decadal or greater) (Fig. 4 and Table 4). The C1-C4 modes exhibit frequent, but generally homogenous oscillations, except during specific periods, as with the Gaira River in 1988-1990, and the Aracataca and

Rancheria Rivers in 1998-2000, when these modes exhibited large oscillations (Fig. 4). In most rivers, the C5 oscillation mode became more pronounced and recurrent about 1995. A similar pattern occurs with the C6-C8 oscillation modes, with the notable exception of the Gaira River, which exhibits quasi-steady, large oscillations on these modes during the entire record. The Aracataca River constitutes the most representative example of such changes (Fig. 4). Residuals show increasing trends



in the Aracataca, Frío, Palomino and Ranchería Rivers, whereas a decreasing trend is seen in the Fundación and Gaira Rivers. Most of these trends showed inflection points in the 1990s or in the earlies 2000s (Fig. 4).

The contribution of modes C1 to C4 (intra-annual to quasi-biennial) ranges between 43.6% and 83.8% (Table 4). These modes provide the highest proportion of the energy spectrum in all rivers, except the Aracataca. In the latter case, the highest proportion of energy comes from modes associated with low-frequency oscillations (i.e. ≥108.1 months), whose contribution amounts to 51.4%. In other rivers, the contribution of low-frequency modes (C6-C8) (i.e. ≥94.1 months) is higher than 12.3%, except in the Ranchería River, where the contribution of these modes amounts to only 3.6%. The contribution of modes associated with the inter-annual signal (C5, except the Aracataca River) varies between 7.4% and 17.9% (Table 4). Discrimination of the contribution of each MFI mode shows that mode C1 (3.5-3.9 months) provides the greatest amount of energy in the Fundación, Frío, Gaira and Ranchería Rivers (21.1-28.0%). Modes C2 (8.1 months) and C5 (108.1 months) provide the highest proportion of energy in the Palomino (19.9%) and Aracataca (19.7%) Rivers. The contribution of modes associated with the quasi-decadal oscillations (94-144 months) is higher than 10.0% in all rivers, except the Ranchería, in which it reaches only 3.6%. The largest contribution of these modes is recorded in the Aracataca River, with 36.1% (Table 4).

The application of the Hilbert-Huang-Transform to each IMF component provides information on amplitude changes at each of the estimated time scales (Fig. 5). High-frequency components (AMP1 - AMP4) show recurrent oscillations in amplitude, which coincide with periods of extreme flow events (Fig. 1 and 5). The inter-annual component (AMP5) showed a marked increase since the 1990s in the Fundación, Aracataca, Frío and Ranchería Rivers, whereas the Gaira and Palomino Rivers exhibited the opposite behavior. The low-frequency components (AMP6 or above) also experienced an increase in amplitude for the same time period, with the exception of the Gaira River and the Fundación River (AMP7). In most of these rivers the amplitude of low-frequency components (≥ AMP6) is comparable, or even higher, to the amplitude exhibited by the inter-annual component (AMP5) (Fig. 5).

### 4.3 Spectral Correlation with Atmospheric/Oceanographic Processes

The Cross Wavelet Transform (XWT) and the Wavelet Coherence (WTC) spectrum show that the AMO, PDO and TNA are correlated and coherent with river streamflow over a range of time scales (Fig. 6-8). The XWT spectrum reveals that AMO and river streamflow exhibited high common powers in bands higher than 64 months, with the highest scale in the ≥96-month bands (≥8 yr). These common powers showed different phase relationships and duration (Fig. 6). For example, for the 96-month band, the XWT revealed a lagged (with AMO leading), but significant common power during the periods 1982-1998 and 1992-2005 in the Fundación and Frío Rivers, respectively. The XWT also revealed an anti-phase significant power in the Aracataca River from 1996 to 2015. Other rivers exhibited very localized significant common powers at high-frequency bands, but not at low-frequency bands (Fig. 6). The WTC also revealed significant coherences with AMO in the low-frequency bands with the Frío, Gaira, Palomino and Ranchería Rivers. Such coherences, however, exhibited a variety of lengths and phase relationships. The Frío, Gaira and Palomino Rivers showed coherence in the ~96-month band throughout the entire record,




with lagged (with AMO leading), and almost in-phase relationships. The Palomino River exhibited a significant in-phase coherence between 1965 and 2002 (Fig. 6). Significant coherences were also observed in other frequency bands. The WTC spectra revealed significant lagged (with AMO leading) coherences in the ~32-month band during the periods 1984-2015 (Gaira River) and 1985-2005 (Ranchería River). The spectra for the Fundación and Palomino Rivers also revealed significant

coherences at the ~32-month band during highly localized periods (Fig. 6).

The XWT analysis revealed significant correlations between PDO and streamflow, particularly in the low-frequency bands (Fig. 7). This relatively high scale-dependence was significant from the early 1990s in most of the rivers, except the Palomino. The Fundación, Aracataca and Frío Rivers experienced significant anti-phase coherences at the ~96-month band from the early 1990s. The Gaira and Ranchería Rivers showed significant lagged (with PDO leading) coherences during the 1985-2005 and

1992-2000 periods, respectively. Patterns obtained through the WTC spectrum were similar (Fig. 7). A period of significant anti-phase coherence between the PDO and streamflow, extending from 1977 to 2015 in the Fundacion River, 1985 to 2015 in the Aracataca River, and 1974 to 2010 in the Frío River, were identified at the ~96 to 128-month bands. The WTC analysis also detected a statistically significant coherent and anti-phase streamflow relationship for the entire record at the ~128-month band in the Gaira River and at the 64-to-128-month bands in the Ranchería River, from 1989 to 2009. Both the XWT and the

WTC spectrum revealed dispersed, significant common powers/coherences at the high-frequency bands. There is a significant coherence observed in most of the rivers at the 32-month band, from 2005 (Fig. 7).

Significant statistical relationships between TNA and streamflow were observed at low-frequency bands (Fig. 8). They were, however, relatively low and dispersed compared to those obtained through the spectral analysis of AMO and PDO (Fig. 6 and 7). The XWT revealed a significant and lagged (with TNA leading) common power in the ~96-month band during the 1975-

2005, 1982-2008 and 1982-1998 periods in the Fundación, Frío and Ranchería Rivers, respectively. The Palomino exhibited high-scale dependence in the ≥64-month bands. Such common powers were mostly not significant, except during a very narrow band in a short period between 1985 and 1995 (Fig. 8). Most of the significant coherences at low-frequency bands fell within the edge effects area, i.e. outside the COI. The Frío, Palomino and Ranchería Rivers exhibited spectral coherences through the entire spectrum at the ≥96-month bands, all significant at the beginning and end of the spectrum, but mainly outside the COI.

The WTC spectrum also revealed highly localized and dispersed significant coherences at the high-frequency bands in most of the rivers, particularly from 1995 to 2015 (Fig. 8). The WTC spectrum showed different phase relationships within each spectral analysis, indicating phase-unlocked coherence (Fig. 8).

## 5 Discussion

### 5.1 Role of Low-Frequency Oscillations in Hydrological Variability

Two approaches indicate that low-frequency oscillations play a significant role in the hydrological variability experienced by rivers of the Sierra Nevada de Santa Marta (Fig. 3-5, and Table 4). The Global Wavelet Spectrum shows that these oscillations



(≥ 8-12 yr) constitute at least a second-order variability source in these rivers, surpassed in some cases only by oscillations associated with the annual band (Fig. 3). The HHT indicates that the contribution to the global energy spectrum from the low-frequency modes is > 12%, reaching up to 51% in the Aracataca River. This contribution is of the same order, or even greater, than the contribution from the inter-annual mode (Table 4). The HHT analysis also revealed that the amplitude of the low-frequency components is of comparable magnitude to, or even greater than the amplitude of the inter-annual components (Fig. 5). Recent studies underscore the importance of quasi-decadal signals on streamflow variability of rivers in northwestern South America (Restrepo et al., 2014). The magnitude of these contributions, however, had not been quantified before, as previous studies in this region focused on assessing the effect of ENSO on hydrological variability (Hastenrath, 1990, Gutierrez and Dracup, 2001, Poveda et al., 2001, Restrepo and Kjerfve, 2004). These studies were also characterized by a strong bias on Andean and Pacific rivers where the influence of ENSO on hydrological variability is predominant (Poveda, 2004; Poveda et al., 2001). It is valid to assume that because of their location, i.e. proximity to the Caribbean Sea, direct exposure to the trade winds and the North Jet Stream, extension (i.e. small drainage basins with limited capacity to filter hydro-climate signals, and low basin storage capacity), and high relief (i.e. favors deep convection), rivers in the SNSM are exposed to greater influence from other climate and atmospheric drivers, particularly variations in sea level pressure (SLP) and sea surface temperature (SST) of the Caribbean Sea (Enfield and Alfaro, 1999).

Our results underscore the role of overlapping oscillations in generating extreme streamflow rates, as noted in other studies (Labat, 2008; Brabets and Walvoord, 2009; Rood et al., 2016; Valdez-Pineda et al., 2017), especially during periods in which the low-frequency oscillations exhibited their maximum power (1998-2002 and 2008-2012) (Fig. 1, 2, 4 and 5). This indicates that the superimposition of different frequency signals can lead to the attenuation or intensification of the hydro-climate signal, depending on the phase of the different oscillatory components. Overlapping oscillations can also lead to a time shift. Low-frequency oscillations seem to play an important role in the intensification (attenuation) of the annual and inter-annual signals in SNSM rivers. The Frío and Aracataca Rivers experienced maximum peak streamflows in 1998 (59.4 $m^3$ $s^{-1}$) and 2002 (97.6 $m^3$ $s^{-1}$), respectively. These values are between 4 and 5 times the inter-annual monthly average (Fig. 1). During those periods, these rivers exhibited simultaneous high-power oscillations in the semi-annual, annual, inter-annual and quasi-decadal bands (Fig. 2 and 4). On the contrary, between 1969 and 1970, when low-frequency oscillations exhibited low strength, and high-frequency oscillations exhibited relatively high power (Fig. 2 and 4), the maximum recorded streamflow was just 38.9 $m^3$ $s^{-1}$ in the Frío River and 62.3 $m^3$ $s^{-1}$ in the Aracataca River (Fig. 1). These represent substantially lower peaks compared to high values observed after the 1990s. Other results also highlight the amplification/attenuation effect of low-frequency oscillations, particularly on the inter-annual signal. In most rivers, the inter-annual signal intensified after 1995, which coincides with an increase in low-frequency oscillations (Fig. 2 and 4). The maximum intensity of the inter-annual signal, which occurred between 1998 and 2002 in most rivers, also coincides with the interval of greater intensity of the quasi-decadal signal (1998-2005) (Fig. 2). Streamflow rates also exhibit inflection points in their trends between the 1990s and 2000s, a period that also coincides with the increase in the amplitude of low-frequency oscillations (Fig. 4).





## 5.2 Role of major oceanographic/atmospheric drivers

Low-frequency oscillations were identified in this study as a source of significant streamflow variability in SNSM rivers (Fig. 2-5 and Table 4). Such oscillations might be associated with climatic/oceanographic drivers, for which modes of variability include quasi-decadal or higher oscillations. The PDO, AMO and TNA are among the most important factors that influence long-term streamflow variability in these rivers (Fig. 6-8 and Supplementary Material). These indices exhibited relatively high spectral correlations (WTC and XWT) with streamflow starting in the 1980s, particularly in the low-frequency bands (≥96 months). Spectral correlations for PDO were more intense and steady throughout the entire SNSM region (Fig. 6), whereas the spectral correlations for AMO and TNA were relatively lower and dispersed, also showing differences between rivers from the western and eastern slopes (Fig. 6 and 7). Maximum power for these spectra occurred in periods during which there were phase changes for such indexes. For example, between 1995 and 2002, when all main oscillatory components exhibited high power and large oscillations (Fig. 2 and 4-5), the AMO and TNA shifted from a negative phase to a positive phase, and the PDO shifted from a warm phase to a cold phase (Fig. 9). These results suggest a relation between changes of these climatic/oceanographic indexes and in long-term streamflow variability, indicating that these watersheds are sensitive to changes in the background climate state. Differences in spectral correlations between rivers from the western and the eastern slopes, and observed differences in phase relationships, indicate that further research is required to draw conclusions about the specific drivers of low-frequency variability.

Studies of the effects that these phenomena (PDO, AMO and TNA) have on hydroclimatology of the Caribbean and South America are relatively recent (Robertson and Mechoso, 1998; Enfield and Alfaro 1999, Tootle et al. 2008, Labat, 2010, Arias et al., 2015, Córdoba-Machado et al., 2016, Valdes-Pineda et al., 2017). Although robust hypotheses have been put forth regarding the physical relation between the PDO (Poveda, 2004), the AMO (Arias et al., 2015) and the TNA (Enfield and Alfaro, 1999) and the climate of northwestern South America, the physical mechanisms by which these phenomena influence the hydrology at low-frequency scales remains elusive. We believe these mechanisms may relate to SST gradients between the Pacific and Atlantic oceans. According to Enfield and Alfaro (1999), anomalous Atlantic SSTs might not be sufficient to promoting hydrologic anomalies when the Pacific is also warm, as occurs during ENSO warm phases. They concluded that opposite SST anomalies in the Tropical North Atlantic and the Eastern Pacific are linked to increased precipitation and streamflow variability over northwestern South America and the Caribbean. Anomalously high streamflows in northwestern South America are promoted by strengthened northeast trade winds, accelerated cross-flow over the Caribbean, and strong SST gradients between the eastern Pacific (Low) and Tropical North Atlantic (High). Since the mid-1990s, the Atlantic Ocean and the Tropical North Atlantic have been warmer (Fig. 9), favouring the occurrence of such rainfall-enhancing mechanisms. These circulation features reflect the southward displacement of the ITCZ and positive SOI anomalies, which in turn are strengthened by SST-positive gradients along the Tropical North Atlantic and eastern Pacific. Such a physical link provides a reasonable explanation for the process of amplification/attenuation of streamflow revealed by spectral analysis. The possible



connection between this low-frequency mode of variability and external forcings, and their influence on a regional scale, warrants further analysis.

Previous studies have shown extremely low correlations between low-frequency phenomena, such as PDO, AMO, and TNA, and streamflow variability of rivers in northwestern South America, suggesting minimal effects on regional hydrology (Gutierrez and Dracup, 2001; Tootle et al., 2008; Córdoba-Machado et al., 2016). Those studies, however, used (1) databases with a strong bias towards rivers of the Pacific and Andean regions, where the influence of ENSO is dominant, (2) databases with no rivers in the SNSM, (3) data on SST primarily from the southern Atlantic Ocean, thereby leaving out regions covered by the AMO, PDO and TNA, (4) estimates of seasonal averages for climatic/oceanographic indices, thus reducing amplitude anomalies, especially for low-frequency oscillations, and (5) linearity for time-series analysis, which is not entirely suitable for detecting hidden signals in non-stationary data such as streamflow and climatic/oceanographic indices. Our study, on the other hand, highlights the significant role of low-frequency oscillations in the hydrological variability of rivers from the SNSM and their potential linkage with large-scale phenomena such as the AMO, PDO, and TNA. Studies using similar approaches have also found such relationships in other regions of South America (Labat et al., 2005; Pasquini and Depetris, 2007; Labat, 2010; Restrepo et al., 2014; Valdes-Pineda et al., 2017).

## 6 Conclusions

Low-frequency oscillations ($\geq$ 8-12 yr) play a significant role in the hydrological variability of rivers in the SNSM. These oscillations did not just exhibit an increase in amplitude, but also became more pronounced and recurrent after about 1995. In most of the studied rivers, the amplitude of low-frequency components was comparable to, or even higher than the amplitude exhibited by the inter-annual component. Low-frequency oscillations constitute at least a second-order variability source in these rivers, surpassed in some cases only by oscillations associated with the annual band. Although intra-annual to quasi-biennial modes provide the highest proportion of the global energy spectrum in all rivers (43.6-83.8%), the contribution from low-frequency modes are > 12% and reach 51% in the Aracataca River.

Periods of intense hydrological variability, in which extreme flows occurred, such as those experienced in 1988-1989, 1998-2000 and 2010-2011, were characterized by the simultaneous occurrence of relatively high-power signals, including low-frequency bands. In addition, the strengthening of the inter-annual signal after 1995, the occurrence of its maximum intensity between 1998 and 2002, and the occurrence of inflection points in the streamflow trends between the 1990s and 2000s, coincide with the increase in the amplitude of low-frequency oscillations, and with the interval of its greatest signal power (1998-2005). Results suggest that streamflow variability is largely dependent on the modulation of low-frequency oscillations. Overlapping of different frequency signals can lead to intensification or attenuation of the hydro-climatological cycle, depending on the phase of the different oscillatory components.





Low-frequency oscillations identified as a source of significant streamflow variability in the SNSM rivers are associated with climatic/oceanographic drivers, with modes of variability that include quasi-decadal or higher oscillations. The XWT and WTC spectra show that the AMO, PDO and TNA are correlated and coherent with river streamflow at different time scales. These indices exhibited relatively high spectral correlations with streamflow starting in the 1980s, particularly in the low-

frequency bands (≥96 months). Spectral correlations for PDO were more intense and steady throughout the entire SNSM region, whereas the spectral correlations for AMO and TNA were relatively lower and dispersed, showing differences between rivers on the western and eastern slopes. Maximum power for these spectra occurred in periods during which there were phase changes of such indexes, suggesting a link between the shift of these climatic/oceanographic indexes and changes in long-term streamflow variability. The physical link between these indexes and hydrologic variability in northwestern South America

might be related to SST and SLP gradients between the Atlantic and Pacific oceans. The physical connection between this low-frequency mode of variability and external forcings warrants further analysis.

Our study highlights the significant role of low-frequency oscillations on the hydrological variability of rivers in the SNSM and potential linkages with large-scale phenomena such as PDO, AMO and TNA. Further analysis is required, however, to examine the role of watershed properties, such as basin storage, baseflow index and groundwater residence time, in establishing

the relation between low-frequency oscillations and streamflow.

**Data availability**

Raw data is available upon request to the authors. Data availability follows IDEAM (Colombian Hydrology and Metereology Institute) policies on access to hydrological data granted to scientific projects.

**Author contribution**

J.C. Restrepo, J.H. Escobar and N. Hoyos designed the experiments; J.C. Restrepo, A. Higgins and S. Ospino carried them out, including performing codes and statistical analysis. J.C. Restrepo prepared the manuscript with contributions from all co-authors.

**Acknowledgments**

This research was partially funded by a grant from the Inter-American Institute for Global Change Research (IAI) CRN3038, which is supported by the US National Science Foundation (Grant GEO-1128040) and by an internal grant from Universidad del Norte. J Escobar and N Hoyos were partially funded by The Canadian Queen Elizabeth II Diamond Jubilee Scholarships





(QES), a partnership among Universities in Canada, the Rideau Hall Foundation (RHF), Community Foundations of Canada (CFC). The QES-AS is made possible with financial support from IDRC and SSHRC. N Hoyos was partially supported by the Fulbright Visiting Scholar Program.

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



**Table 1.** Drainage basin (A), headwater elevation, mean monthly streamflow (Q), maximum monthly streamflow ($Q_{max}$), minimum monthly streamflow ($Q_{min}$), flood regimes ($Q_{max}/Q$) and discharge variability ($Q_{max}/Q_{min}$) of the rivers draining the Sierra Nevada de Santa Marta.

| River | A (10³ km²) | Headwater (m.a.s.l.) | Q (m³ s⁻¹) | $Q_{max}$ (m³ s⁻¹) | $Q_{min}$ (m³ s⁻¹) | $Q_{max}/Q$ (-) | $Q_{max}/Q_{min}$ (-) |
|---|---|---|---|---|---|---|---|
| Fundación | 1.87 | 2986 | 28.52 | 140.9 | 4.83 | 4.94 | 29.2 |
| Aracataca | 0.93 | 4408 | 18.19 | 97.6 | 0.78 | 5.36 | 125.1 |
| Frío | 0.32 | 3716 | 13.76 | 59.5 | 2.06 | 4.32 | 28.9 |
| Gaira | 0.10 | 2750 | 2.30 | 18.2 | 0.17 | 7.91 | 107.1 |
| Palomino | 0.68 | 4785 | 25.51 | 108.8 | 4.52 | 4.26 | 24.1 |
| Ranchería | 4.23 | 3700 | 12.03 | 121.5 | 0.06 | 10.09 | 2025.0 |



**Table 2.** Information on climate índices used in this study.

| Climatic/ Oceanographic Indices | Definition | Coverage/ Time resolution | Source |
|---|---|---|---|
| Atlantic Meridional Oscillation (AMO) | Simple basin average of North Atlantic Ocean (0° - 70°) Sea Surface Temperatures (SSTs). The AMO index consists of detrended SST anomalies through this Atlantic Ocean region. The AMO displays a low-frequency periodicity of 65-80 years. | 1950 – 2015/ Monthly | Climate Prediction Center – NOAA (NOAA, 2016) |
| Pacific Decadal Oscillation (PDO) | Oceanographic/Atmospheric phenomena associated with persistent, bimodal climate patterns in the northern Pacific Ocean (poleward of 20°N). It oscillates with a characteristic period about 50 years – a particular phase will typically persist for about 25 years. | 1950 – 2015/ Monthly | Climate Prediction Center – NOAA (NOAA, 2016) |
| Tropical North Atlantic (TNA) | Indicator of the surface temperature anomalies in the eastern tropical North Atlantic Ocean. It is calculated with SSTs in the box 55°W - 15°W, 5°N - 25°N. The anomaly is calculated relative to a monthly climatological seasonal cycle based on the years 1982-2005. | 1950 – 2015/ Monthly | Climate Prediction Center – NOAA (NOAA, 2016) |
| Southern Oscillation Index (SOI) | Standardized index based on the observed sea level pressure differences between Tahiti and Darwin, Australia. A measure of the large-scale fluctuations in air pressure occurring between the western and eastern tropical Pacific | 1950 – 2015/ Monthly | Climate Prediction Center – NOAA (NOAA, 2016) |
| Multivariate ENSO Index (MEI) | It is based on the six main observed variables over the tropical Pacific: sea-level pressure, zonal and meridional components of the surface wind, sea surface temperature, surface air temperature, and total cloudiness fraction of the sky. It is computed separately for each of twelve sliding bi-monthly seasons and calculated as the first unrotated Principal Component (PC) of all six observed fields combined. | 1949/1950 – 2015/ Monthly-composite | Climate Prediction Center – NOAA (NOAA, 2016) |
| Tropical South Atlantic Index (TSA) | Indicator of the surface temperature anomalies in the Gulf of Guinea, the eastern tropical South Atlantic Ocean. It is calculated with SSTs in the box 30°W - 10°E, 20°S - 0°. The anomaly is calculated relative to a monthly climatological seasonal cycle based on the years 1982-2005. | 1950 – 2015/ Monthly | Climate Prediction Center – NOAA (NOAA, 2016) |
| Caribbean SST Index (CAR) | The time-series of SST anomalies averaged over the Caribbean. Anomalies were calculated relative to the 1951-2000 climatology. | 1950 – 2010/ Monthly-composite | Climate Prediction Center – NOAA (NOAA, 2016) |
| North Tropical Atlantic Index (NAT) | Indicator of the surface temperatures in a broad swath of the tropical North Atlantic Ocean. It is calculated with SSTs in the box 40°W - 20°W, 5°N - 20°N. The anomaly is calculated relative to a monthly climatological seasonal cycle based on the years 1982-2005. | 1950 – 2010/ Monthly | Climate Prediction Center – NOAA (NOAA, 2016) |





**Table 3.** Rivers and gauging stations analyzed in this study. The location and historic record of freshwater discharge data are also included.

| River | Gauging Station | Location | | | Record |
| --- | --- | --- | --- | --- | --- |
| | | Elevation (m.a.s.l.) | Longitude | Latitude | |
| *Sierra Nevada Rivers* | | | | | |
| 1. Fundación | Fundación | 55 | 74°11W | 10°31N | 1958 – 2013 |
| 2. Aracataca | Puente Ferrocarril | 37 | 74°11W | 10°35N | 1965 – 2013 |
| 3. Frío | Rio Frío | 30 | 74°09W | 10°34N | 1965 – 2009 |
| 4.Gaira | Minca | 650 | 74°07W | 11°08N | 1978 – 2013 |
| 5. Palomino | Puente Carretera | 30 | 73°34W | 11°14N | 1965 – 2013 |
| 6. Ranchería | Hacienda Guamito | 76 | 72°37W | 11°10N | 1976 – 2007 |





**Table 4.** Period (T) and percentage of energy for each IFM mode estimated for the rivers of the Sierra Nevada of Santa Marta - contribution to the global energy spectrum.

| IMF Modes | R. Fundación | | R. Aracataca | | R. Frío | | R. Gaira | | R. Palomino | | R. Ranchería | |
|---|---|---|---|---|---|---|---|---|---|---|---|---|
| | T (months) | Energy (%) | T (months) | Energy (%) | T (months) | Energy (%) | T (months) | Energy (%) | T (months) | Energy (%) | T (months) | Energy (%) |
| C1 | 3.8 | 23.3 | 3.8 | 6.6 | 3.9 | 21.1 | 3.6 | 28.0 | 3.4 | 18.9 | 3.5 | 24.3 |
| C2 | 8.8 | 20.9 | 8.2 | 10.3 | 8.7 | 14.0 | 8.0 | 18.7 | 8.1 | 19.9 | 7.3 | 22.4 |
| C3 | 16.8 | 17.2 | 15.4 | 11.4 | 17.0 | 15.4 | 13.2 | 20.2 | 14.9 | 18.6 | 16.0 | 22.8 |
| C4 | 35.9 | 14.1 | 29.4 | 15.3 | 33.4 | 9.6 | 28.7 | 7.8 | 26.0 | 14.1 | 32.2 | 14.3 |
| C5 | 51.7 | 7.4 | 108.1 | 19.7 | 58.1 | 17.9 | 48.6 | 8.4 | 72.7 | 13.3 | 61.6 | 11.2 |
| C6 | 119.2 | 14.4 | 120.7 | 16.4 | 94.1 | 14.9 | 120.4 | 10.0 | 96.1 | 1.9 | 120.3 | 3.6 |
| C7 | 257.3 | 2.4 | 195.5 | 12.7 | 191.8 | 2.5 | 266.1 | 3.7 | 144.2 | 10.4 | - | - |
| C8 | - | - | 229.1 | 2.6 | - | - | - | - | - | - | - | - |




**Figure 1. (A) Map of the different drainage basins, with locations of streamflow stations and major geographic features; (B) Historical monthly streamflow series of the Fundación, Aracataca, Frío, Gaira, Palomino and Rancheria Rivers.**





**Figure 2. Continuous Wavelet transform spectrum for the Fundación, Aracataca, Frío, Gaira, Palomino and Ranchería Rivers. The dark yellow/light blue colours in the wavelet spectra correspond to high/low values of the transform coefficients (power). The thick black contour delimits the 95% confidence level against AR(1) red noise, and the cone of influence where edge effects ($T/\sqrt{2}$) are not negligible is shown as a black line.**






**Figure 3. Global Wavelet spectrum for original and filtered time series of the Fundación, Aracataca, Frío, Gaira, Palomino and Ranchería Rivers.**

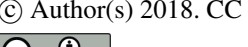


**Figure 4. Ensembled Empirical Mode Decomposition (EEMD) for the Fundación, Aracataca, Frío, Gaira, Palomino and Ranchería Rivers.**



**Figure 4. (Cont.) Ensembled Empirical Mode Decomposition (EEMD) for the Fundación, Aracataca, Frío, Gaira, Palomino and Ranchería Rivers.**





**Figure 5.** Temporal distribution of amplitudes (AMP) (m$^3$ s$^{-1}$) for each component (IMFs) of the Aracataca, Frío, Gaira, Palomino and Ranchería Rivers.



Figure 5. (Cont.) Temporal distribution of amplitudes (AMP) (m³ s⁻¹) for each component (IMFs) of the Aracataca, Frío, Gaira, Palomino and Ranchería Rivers.





**Figure 6. Cross Wavelet Transform (XWT) and Wavelet Coherence (WTC) between AMO and the (A) Fundación, (B) Aracataca, (C) Frío, (D) Gaira, (E) Palomino, and (F) Ranchería Rivers.**



**Figure 7. Cross Wavelet Transform (XWT) and Wavelet Coherence (WTC) between PDO and the (A) Fundación, (B) Aracataca, (C) Frío, (D) Gaira, (E) Palomino, and (F) Ranchería Rivers.**





**Figure 8. Cross Wavelet Transform (XWT) and Wavelet Coherence (WTC) between TNA and the (A) Fundación, (B) Aracataca, (C) Frío, (D) Gaira, (E) Palomino, and (F) Ranchería Rivers.**





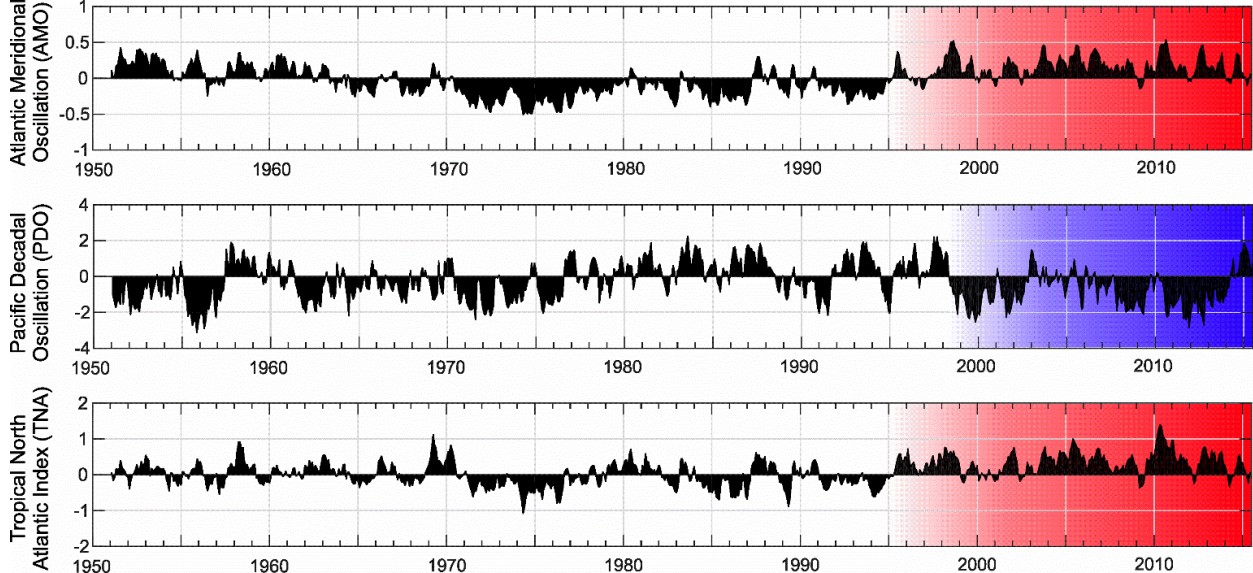

**Figure 9. Monthly values of Atlantic Meridional Oscillation (AMO), Pacific Decadal Oscillation and Tropical North Atlantic Index (TNA) anomalies (1951-2015). Colored boxes highlight the last shift in the phase of these indexes.**