# Peer review of "Contribution of low-frequency climatic/oceanic oscillations to streamflow variability in small, coastal rivers of the Sierra Nevada de Santa Marta (Colombia)"

_Hydrology and Earth System Sciences, 2018_

## Referee Comment (RC1) · Anonymous Referee #1 · 21 Jan 2019

A review of the paper "Contribution of low-frequency climatic/ocean oscillations to stemflow variability in small, coastal rivers of the Sierra Nevada de Santa Marta (Colombia)" by Juan Camilo Restrepo, Aldemar Higgins, Jaime Escobar, Silvio Ospino and Natalia Hoyos.

The present manuscript addresses an important and current subject in Hydrology. The influence of large-scale oceanographic/atmospheric processes on streamflow variability is a research question of high importance in Hydrology. The understanding of how low-frequency oscillations identified in climate indices can drive the variability in the

flow regime in rivers allows us to count on a valuable tool for the construction of statistical models. Spectral analysis was undertaken to determine the nature and magnitude of the relationship between monthly streamflow of 6 rivers and large-scale atmospheric/oceanographic circulation patterns. The study focused on basins that have special characteristics, are small, tropical, coastal mountain rivers localized in Colombia. Continuous wavelet transform and Hilbert Huang transform were the methods selected to identify the modes of variability in the rivers and climatic/oceanographic indices. Cross-wavelet analysis and wavelet coherence that are powerful methods for testing a proposed linkage between two time series were also used by the authors in the paper. The results exhibit that streamflow variability are strong associated with modes of variability in the Atlantic Meridional Oscillation (AMO), Pacific Decadal Oscillation (PDO) and Tropical North Atlantic (TNA).

Due to the complexity that can exist in the teleconnection between climatic indices and flow regime in a river the authors selected the appropriate tools. The tools selected to carry out the study, allows to overcome the problem of linear analysis when evaluating the relationship between low-frequency phenomena and streamflow variability of rivers.

The manuscript is reasonable well-structured, the methods are well described, and the research is within the scope of HESS. However, the manuscript requires a more in-depth discussion of the results and it is necessary to be incorporated some missing important information.

The paper deserves to be published on Hydrological and Earth Science Systems, after some minor changes. I am reporting below some specific comments, which I hope the authors will find useful while revising their manuscript.

Comments:

It's necessary to highlight the novelty of the work because it's no clear. If this work would not be published, what would the international hydrology community miss? Novelty can reside in a new data set which is of importance to the international hydrology

community, in new methodological development, in new conceptual ideas or novel interpretation and insights. The paper applies established methods and it follows the ideas that many papers have developed/applied. The conclusions seem not to add new findings to the already existing knowledge.

I recommend that the authors specify the type of regimen (natural or altered) in the flow gauge stations. This point is highly important for the results.

It's important to know if the flow gauge stations are in the upper, middle or lower part of the basins. I recommend that the authors should incorporate the spatial location of flow gauging stations in Figure 1A.

Wavelet power relations and phase relations between monthly streamflow of the rivers and large-scale circulation patterns are relatively stable in the longer periods (> 2 years band) and are very unstable in the shorter periods (< 2 year band). This can demonstrate that from longer periods, the monthly streamflow could be controlled by the slowly changing climate. During shorter periods, the monthly streamflow is not only controlled by large-scale ocean–atmosphere patterns.

One point that is not discussed in depth in the results is the phase changes in the relationship between the time series of flows and the climatic indices. The phase relationship between climatic indices and streamflow is changing in shorter and longer periods. The different phase relationships between AMO, TNA and PDO and monthly streamflow could be show the different influences of variables of the atmospheric system.

It's necessary and very helpful for readers to indicate in the cross-wavelet transform and squared wavelet coherence that the relative phase relationship is shown by dark arrows.

Due to the short length of the flow gauge stations records, it is risky to explore the statistical presence of decadal oscillations. Specifically, the variability mode C8, which

do not seem to have enough statistic evidence.

Page 3, line 23: ". ....which was designated a RAMSAR site because. ...." What is the meaning of RAMSAR?

Page 5, line 34: "Data series with a non-normal distribution were transformed prior to applying . . . . . ." What type of transformation was used?

The research results of this paper present apparent opportunities for improving forecasting of streamflow along the coastal rivers of the Sierra Nevada de Santa Marta, which, in turn, will improve water resources management.

––––––––––––––––––––––

---

## Referee Comment (RC2) · Anonymous Referee #2 · 1 Feb 2019

This study evaluates the influence of low-frequency oscillations linked to large-scale oceanographic-atmospheric processes, on streamflow variability in small tropical coastal mountain rivers of the Sierra Nevada de Santa Marta, Colombia. By using spectral analysis and Hilbert Huang transform, the study aims to (1) explore temporal characteristics of streamflow variability, (2) estimate the net contribution to the energy spectrum of low-frequency oscillations to streamflow anomalies, and (3) analyze the linkages between streamflow anomalies and large-scale, low-frequency oceanographic/atmospheric processes.

[Figure]

The main topic of the article is important to Hydrology and water resource management, and deserves to be published in HESS. However, the results need to be discussed in a broader context, comparing the main findings with related literature. The tools applied to address the research questions are adequate and properly applied, however some technical details are necessary to be described. In addition, a deeper explanation about the physical mechanisms linking PDO, AMO, TNA and the basins' hydrology is necessary. Also, the whole subject is about the possibility of a cause-effect relation between decadal oscillations and streamflow, but the concept of phase locked signals is completely missing in the interpretation of the results and the discussion, which I think is necessary. Thus, my decision is accepted with major revisions.

Specific comments:

- First paragraph: a more in-depth description on the PDO-ENSO relation is necessary in addition to AMO and TNA relations to inter-annual oscillations.

- Second paragraph: the main idea is confusing. Maybe split paragraphs one for novel statistical methods and another related to the hydrology in Colombia.

- Third paragraph: to keep the logic of the manuscript the main objectives ought to be aligned with the sub-sections presented in section 4.

- Pag. 4 line 30: explain the main difference between XWT and WTC.

- Pag. 7 line 10: equation (7) may be wrong.

- Pag. 8 line 6-7: from Fig. 2 the statement is not evident for station Frío, please explain.

―――――――――――――

---

## Author Comment (AC1) · 1 Mar 2019

Anonymous Referee #1 A review of the paper "Contribution of low-frequency climatic/ocean oscillations to stemflow variability in small, coastal rivers of the Sierra Nevada de Santa Marta (Colombia)" by Juan Camilo Restrepo, Aldemar Higgins, Jaime Escobar, Silvio Ospino and Natalia Hoyos.

The present manuscript addresses an important and current subject in Hydrology. The influence of large-scale oceanographic/atmospheric processes on streamflow

variability is a research question of high importance in Hydrology. The understanding of how low-frequency oscillations identified in climate indices can drive the variability in the flow regime in rivers allows us to count on a valuable tool for the construction of statistical models. Spectral analysis was undertaken to determine the nature and magnitude of the relationship between monthly streamflow of 6 rivers and large-scale atmospheric/oceanographic circulation patterns. The study focused on basins that have special characteristics, are small, tropical, coastal mountain rivers localized in Colombia. Continuous wavelet transform and Hilbert Huang transform were the methods selected to identify the modes of variability in the rivers and climatic/oceanographic indices. Cross-wavelet analysis and wavelet coherence that are powerful methods for testing a proposed linkage between two time series were also used by the authors in the paper. The results exhibit that streamflow variability are strong associated with modes of variability in the Atlantic Meridional Oscillation (AMO), Pacific Decadal Oscillation (PDO) and Tropical North Atlantic (TNA).

Due to the complexity that can exist in the teleconnection between climatic indices and flow regime in a river the authors selected the appropriate tools. The tools selected to carry out the study, allows to overcome the problem of linear analysis when evaluating the relationship between low-frequency phenomena and streamflow variability of rivers.

The manuscript is reasonable well-structured, the methods are well described, and the research is within the scope of HESS. However, the manuscript requires a more indepth discussion of the results and it is necessary to be incorporated some missing important information. The paper deserves to be published on Hydrological and Earth Science Systems, after some minor changes. I am reporting below some specific comments, which I hope the authors will find useful while revising their manuscript.

Authors: We appreciate your comments and the overall assessment of the work. It does contribute to improve the quality of the manuscript. Below we provide answers to your specific comments as Author's Response (AR) and Author's Changes in

Manuscript (ACM).

Comments from referees:

(1) It's necessary to highlight the novelty of the work because it's no clear. If this work would not be published, what would the international hydrology community miss? Novelty can reside in a new data set which is of importance to the international hydrology community, in new methodological development, in new conceptual ideas or novel interpretation and insights. The paper applies established methods and it follows the ideas that many papers have developed/applied. The conclusions seem not to add new findings to the already existing knowledge.

(AR) Studies that look at the interplay of multiple atmospheric/oceanographic oscillations on streamflow long-term variability are relatively recent (eg, Shi et al., 2017; Su et al., 2018; Murgulet et al., 2018). The lack of studies on this subject has led to an increase in research looking at how multiple large-scale climatic/oceanographic oscillations, particularly low-frequency, drives streamflow variability (eg, Tootle et al., 2008, Yang et al., 2011, Massei and Fournier, 2012, Boers et al., 2014, Córdoba-Machado et al., 2016, Schulte et al., 2016, Shi et al., 2017; Su et al., 2018; Murgulet et al., 2018). Thus, neither the approach nor the methods applied are novel or unique, since they have been widely applied during the last years. The novelty of this study lies, fundamentally, in the location (Caribbean) and basins' physiography from which the analyzed data come from (Page 2- Line 31, Page 3 - Line 1). These unique charactersitics allowed us to: (1) highlight the influence of low frequency climate indices (ie PDO, AMO and TNA) on the surface hydrology of northern South America (where its effect had previously been minimized - Page 13 Line 1) (2) provide strong evidence that low frequency oscillations are major players on the streamflow variability for this area. Its magnitude is of the same order (or higher in some cases) than that of the ENSO (considered as the main driver of the superficial hydrology of Northwestern Southamerica) (Page 11 Line 3, Page 11 Line 7) (3) show that flow variability is a consequence of the concurrence of different frequency signals, rather than to a specific signal (Labat,

2008, Brabets and Walvoord, 2009, Rood et al., 2016, Valdez-Pineda et al., 2017). This also highlights the modulating effect of quasi-decadal signals (Page 11 Line 16) (4) understand the role of low frequency oscillations in the streamflow of basins with a small drainage area. These signals had shown less intenisty in regional scale studies. (i.e. Murgulet et al., 2017) (Page 11 Line 11). All these aspects will be highlighted explicitly in the introduction and conclusions.

(AMC) In order to highlight the novelty of the work, the manuscript will be modified as follows (new text in bold and cursive):

"(Page 3 – Line 12) To our knowledge, this is the first study to estimate the contribution of low-frequency oscillations to the hydrologic variability at a subregional scale and in these type of watersheds (i.e. small, coastal, and mountainous), and specifically in northern South America, where ENSO has been identified previously as the preeminent driver on streamflow variability (i.e. Gutierrez and Dracup, 2001; Poveda et al., 2001; Córdoba-Machado et al., 2016)".

"(Page 13 – Line 17) Low-frequency oscillations ($\geq$ 8-12 yr) play a significant role in the hydrological variability of rivers in the SNSM (…) In most of the studied rivers, the amplitude of low-frequency components was comparable to, or even higher than the amplitude exhibited by the inter-annual component, which has been considered previously as the main driver of streamflow variability in northern South America at a regional scale. Low-frequency oscillations constitute at least a second-order variability source in these rivers, surpassed in some cases only by oscillations associated with the annual band. Although intra-annual to quasi-biennial modes provide the highest proportion of the global energy spectrum in all rivers (between 43.6 and 83.8%), the contribution from low-frequency modes are > 12%, and reach up to 51% in the Aracataca River, indicating an active effect of such low-frequency oscillations in the streamflow variability at a subregional scale in northern South America. Such effect deserves further studies."

"(Page 14 – Line 1) Previous studies have shown a very low correlation between low-frequency phenomena and streamflow variability in northwestern South America, suggesting minimal effects on regional hydrology. The sub-regional scale approach and the statistical spectral analysis of this study allow to identify and estimate a significant contribution of low-frequency oscillations in the streamflow variability of the SNSM Rivers. Such oscillations, identified as a source of significant streamflow variability in the SNSM rivers, are associated with large-scale climatic/oceanographic drivers, with modes of variability that include quasi-decadal or higher oscillations. The XWT and WTC spectra show that the AMO, PDO and TNA are correlated and coherent with river streamflow at different time scales (. . .) suggesting a link between the shift of these climatic/oceanographic indexes and changes in long-term streamflow variability.

"(Page 13 – Line 24) Periods of intense hydrological variability, in which extreme flows occurred, such as those experienced in 1988-1989, 1998-2000 and 2010-2011, were characterized by the simultaneous occurrence of relatively high-power signals, including low-frequency bands (. . .) Overlapping of different frequency signals can lead to intensification or attenuation of the hydro-climatological cycle, depending on the phase of the different oscillatory components. These pattern highlights the importance of the interaction of different frequency signals and their phase shifting interactions on the streamflow variability of these rivers."

"(Page 14 – Line 12) Our study highlights the significant role of low-frequency oscillations on the hydrological variability of rivers in the SNSM and potential linkages with large-scale phenomena such as PDO, AMO and TNA. We hypothesize that the location and the physiography of these watersheds (i.e. proximity to the Caribbean Sea, direct exposure to the trade winds and the North Jet Stream, small drainage basins, low basin storage capacity and high relief) make rivers more exposed to sea level pressure (SLP) and sea surface temperature (SST) anomalies; particularly, from the Atlantic Ocean and the Caribbean Sea. Further work is necesary to examine the role of these watershed properties, and others such as basin storage, baseflow index and groundwater residence time, in establishing the relation between low-frequency oscillations and streamflow variability."

(2) I recommend that the authors specify the type of régimen (natural or altered) in the flow gauge stations. This point is highly important for the results.

(AR) It can be argued that for the surveyed period (Table 3) these rivers experienced a quasi-natural hydraulic regime since there was no damming or major hydraulic structures on the riverbeds. Construction of the only dam located in the study area (Ranchería river) began in the late 2000's decade and our anlysis do not include this time interval. These basins have experienced, however, significant land-use changes (mainly deforestation and an increase in agriculture). Land-use changes have limited basin hydrologic modulation capacity and favoured changes in hydrological patterns (i.e. ocurrence of extreme events, seasonality length and intensity).

(AMC) In order to clarify the type of regime depicted in the streamflow time series, the manuscript will be modified (Page 3 – Line 24) as follows (new text in bold and cursive):

"In addition, the SNSM rivers exhibit high to very high discharge variability (Qmax/Qmin), high flood regime (Qmax/Q), while possessing drainage areas < 5.0 x103 km2 in mountainous zones (Table 1). Thus, topography is a primary factor controlling flood variability (Restrepo et al., 2014). Except for the Ranchería River with a dam built in the late 2000's decade, the SNSM rivers have no damming or fragmentation. However, just 15% of the natural forest remains completely unaltered, due to widespread logging and and incease in agriculture. Only 8.5% of the river headwaters remain pristine (Fundación Pro-Sierra Nevada de Santa Marta, 1997). These land-use changes have led to a general loss of hydrologic modulation capacity in the watersheds, which in turn have favoured the occurrence of changes in the hydrological patterns; specifically, an increase of seasonal streamflow extremes (e.g. Pierini et al., 2017; Hoyos et al., 2019)".

(3) It's important to know if the flow gauge stations are in the upper, middle or lower

part of the basins. I recommend that the authors should incorporate the spatial location of flow gauging stations in Figure 1A.

(AR) Gauge stations were located in the lower part of the basin, closest to the river mouth. Streamflow time series measured at the mouth of the watershed is considered a valuable integrated signal for drainage basin's water cycle (i.e., precipitation, evapotranspiration, runoff) (e.g., Garcia and Mechoso, 2005; Milliman et al., 2008; Labat, 2010; Pasquini and Depetris, 2007; Probst and Tardy, 1987; Restrepo et al., 2014). Location of the gauge stations will be added to Figure 1A.

(AMC) In order to clarify the gauging stations location, the manuscript and Figure 1 will be modified (Page 4 – Line 19) as follows (new text in bold and cursive):

"Selection of streamflow gauging stations was based on the location and length of records, which had to be sufficiently long to enable analysis of the role and properties of low-frequency oscillations on streamflow variability. Gauge stations are located close to the river mouth, in the lower part of the basin. Streamflow time series measured close to the river mouth are considered a reliable integrated signal for drainage basin's water cycle (e.g., Garcia and Mechoso, 2005; Milliman et al., 2008; Labat, 2010; Pasquini and Depetris, 2007; Probst and Tardy, 1987; Restrepo et al., 2014). . .".

(4) Wavelet power relations and phase relations between monthly streamflow of the rivers and large-scale circulation patterns are relatively stable in the longer periods (> 2 years band) and are very unstable in the shorter periods (< 2 year band). This can demonstrate that from longer periods, the monthly streamflow could be controlled by the slowly changing climate. During shorter periods, the monthly streamflow is not only controlled by large-scale ocean–atmosphere patterns.

(AR) This is mentioned on the results section (Page 9 Line 30, Page 10 – Line 14, Page 10 – Line 25). We will highligt this statement in the discussion.

(AMC) In order to highlight the differences in the phase relationship when comparing

high and low frequencies, the manuscript will be modified as follows (new text in bold and cursive):

Results (Page 9 – Line 24): "The Cross Wavelet Transform (XWT) and the Wavelet Coherence (WTC) spectrum show that the AMO, PDO and TNA are correlated and coherent with river streamflow over a range of time scales and frequencies. Differences are noticeable, however, for power and phase-relationship when comparing high- and low-frequency signals (Fig. 6-8)".

Discussion (Page 12 – Line 13): "These results suggest a relation between changes of these climatic/oceanographic indexes and long-term streamflow variability, indicating that these watersheds are sensitive to changes in the background climate state. Furthermore, power and phase relationships between streamflow and different índices (Fig. 6-8) were relatively steady for low-frequencies (i.e. > 96 months) but unstable and disperse for high-frequencies (i.e. < 96 months). Such difference in patterns, suggest that during longer periods, streamflow might be modullated by the slowly change in the climate background climate; whereas during shorter periods, the streamflow is not only controlled by large-scale ocean–atmosphere patterns, but also by local short-term phenomena. This result highlights, once again, the significant effect of the superposition of signals of different frequencies in the streamflow variability (eg Pasquini and Depetris, 2007, Labat, 2010, Steinman et al., 2014, Shi et al., 2016 Murgulet et al., 2017)".

(5) One point that is not discussed in depth in the results is the phase changes in the relationship between the time series of flows and the climatic indices. The phase relationship between climatic indices and streamflow is changing in shorter and longer periods. The different phase relationships between AMO, TNA and PDO and monthly streamflow could be show the different influences of variables of the atmospheric system.

(AR) This is correct. This aspect is now discussed in more detail in the discussion

section. It is important to keep in mind, however, that the understanding and discussion of the physical links between streamflow variability and these large-scale phenomena, is beyond the scope and aim of this study (Page 3 - Line 8) . Following your suggestions will adjust the discussion.

(AMC) In order to discuss broadly the phase relationship between streamflow and climatic/oceanographic indices, the manuscript will be modified as follows (new text in bold and cursive):

Discussion (Page 12 – Line 13): "These results suggest a relation between changes of these climatic/oceanographic indexes and long-term streamflow variability, indicating that these watersheds are sensitive to changes in the background climate state. Furthermore, power and phase relationships between streamflow and different indices (Fig. 6-8) were relatively steady for low-frequencies (i.e. > 96 months) but unstable and disperse for high-frequencies (i.e. < 96 months). Such differents in these patterns, suggest that during longer periods, the streamflow might be modullated by the slowly change in the climate background state; whereas during shorter periods, the streamflow is not only controlled by large-scale ocean–atmosphere patterns, but also by local short-term phenomena. This result highlights, once again, the significant effect of the superposition of signals of different frequencies in the streamflow variability (e.g. Pasquini and Depetris, 2007; Labat, 2010; Steinman et al., 2014; Shi et al., 2016; Murgulet et al., 2017). For the lower frequencies, in both the XWT and WTC analysis, the phase relationship exhibited a stable phase lag inside the significance common power regions for each river (Fig. 6-8). Such consistent varying phase lag, implies a phase-locked relationship and suggests a physically link (i.e. not a casual relationship) between the streamflow variability and each of the climatic/oceanographic indices (Grinsted et al., 2004; Labat, 2005). Outside areas with significant power the phase relationship changed (Fig. 6-8). We therefore speculate that despite the relatively strong link between streamflow and these indices at specific frequencies (low) and temporal windows (Fig. 6-8), these relationships are highly non-linear and nonstationary; depending heavily on the phase experienced by these oscillations and their dynamic feedback processes (e.g. Battisti and Sarachick, 1995; Einfeld and Alfaro, 1999; Garreaud et al., 2009). Differences in spectral correlations between rivers from the western and the eastern slopes, and differences in phase relationships observed in some rivers, indicate that further research is required to draw conclusions about the specific drivers of low-frequency variability.

(6) It's necessary and very helpful for readers to indicate in the cross-wavelet transform and squared wavelet coherence that the relative phase relationship is shown by dark arrows.

(AR) The type of relative phase relationships depicted in the XWT and WTC analyses (Page 5 – Line 19), as well as their statistical significance (Page 5 – Line 30), were explained in Section 3.2. However, a further indication of the meaning of the dark arrows in the XWT and WTC analyses (in the Figures) will improve the manuscript as well as the reader's ability to interpret results.

(AMC) In order to clarify the meaning of the dark arrows in the XWT and WTC analyses the captions of the Figures 6, 7 and 8 will be modified as follows (new text in bold and cursive):

"Figure 6. Cross Wavelet Transform (XWT) and Wavelet Coherence (WTC) between AMO and the (A) Fundación, (B) Aracataca, (C) Frío, (D) Gaira, (E) Palomino, and (F) Ranchería Rivers. Dark arrows enclosed in the significant regions (thick black contours) represent the angle-phase relationships. For explanation on types and statistical significance of such relationships see Section 3.2".

"Figure 7. Cross Wavelet Transform (XWT) and Wavelet Coherence (WTC) between PDO and the (A) Fundación, (B) Aracataca, (C) Frío, (D) Gaira, (E) Palomino, and (F) Ranchería Rivers. Dark arrows enclosed in the significant regions (thick black contours) represent the angle-phase relationships. For explanation on types and statistical significance of such relationships see Section 3.2".

"Figure 8. Cross Wavelet Transform (XWT) and Wavelet Coherence (WTC) between TNA and the (A) Fundación, (B) Aracataca, (C) Frío, (D) Gaira, (E) Palomino, and (F) Ranchería Rivers. Dark arrows enclosed in the significant regions (thick black contours) represent the angle-phase relationships. For explanation on types and statistical significance of such relationships see Section 3.2".

(7) Due to the short length of the flow gauge stations records, it is risky to explore the statistical presence of decadal oscillations. Specifically, the variability mode C8, which do not seem to have enough statistic evidence.

(AR) Yes, we agree. The length of the time series determines the reliability of the analysis, as well as the temporal length of the information that can be obtained from them. For this reason, it was indicated that in order to obtain information on quasi-decadal oscillations, the series evaluated should have a minimum extension of 32 years to comply with the requirements of edge effects ($T/2\sqrt{2}$) and cutoff frequency ($T/2$) approaches (Page 4 - Line 21). The time series evaluated have lengths that oscillate between 32 and 55 years (Table 3). We can then obtain statistically significant information about oscillations ranging up to 11.3-19.4 yrs and 16-27 yrs, according to the edge effect and cutoff frequency approach, respectively. These time series provide reliable information for all rivers evaluated in terms of the statistical presence of decadal oscillations. It is true also that from these time series it might be risky to explore the presence of higher period oscilations, particularly to those corresponding to the larger MFI modes in the Table 4. This is the specific case for River Fundacion (C7: 21 years) and Gaira (C7: 22 years) (see Table 4). This last aspect needs to be clarified in the manuscript.

(AMC) In order to clarify aspects linked to the presence and the statistical significance of larger period oscillations within the streamflow time series analyzed, the manuscript will be modified as follows (Page 8 – Line 28) (new text in bold and cursive):

"Mode C6 and higher modes correspond to low-frequency oscillations (i.e. quasi-decadal or greater) (Fig. 4 and Table 4). Information on the last IMF mode of Fundación (C7) and Gaira (C7) Rivers must be analyzed cautioulsy as they are outsied the range established for the edge effects approach (Table 4)."

(8) Page 3, line 23: ".....which was designated a RAMSAR site because....." What is the meaning of RAMSAR?

(AR) In 1971 the Ramsar Convention was first organized to build an international treaty to provide the framework for national action and international cooperation for the conservation and wise use of wetlands and their resources. This Convention considered granting Ramsar status to those wetlands of international importance due to their biological wealth and their role as shelters for seasonal migratory waterbirds (Ramsar, 2019). More than 160 countries have ratified this Convention, and thus it is widely recognized worldwide.

(ACM) In order to clarify the specific meaning of RAMSAR site, the manuscript will be modified (Page 3 – Line 23) as follows (new text in bold and cursive):

"...Rivers that run through the western slopes of the SNSM flow into the Ciénaga Grande de Santa Marta (CGSM), the largest Colombian coastal lagoon (∼730 km2) (Fig. 1), which was designated a Ramsar wetland because of its ecological diversity and importance and its role as a shelter for migratory birds (Ramsar, 2019)."

(9) Page 5, line 34: "Data series with a non-normal distribution were transformed prior to applying ......" What type of transformation was used?

(AR) Many statistical analyzes assume that data is normally distributed. Our preliminary analysis showed, however, that the flow series used in this work did not conform to a normal distribution. Therefore, the Wavelet analysis needed a transformation of the probability density functions for the time series to generate reliable results. The transformation of the data consisted of its standardization, the calculation of a zero mean and an unit standard deviation. These are widely established procedures (e.g. Torrence and Compo, 1998, Grinsted et al., 2004, Labat, 2005).

(ACM) In order to clarify the data transformation process, the manuscript will be modified (Page 5 – Line 2) as follows (new text in bold and cursive):

"Data series with a non-normal distribution were transformed prior to applying the CWT, XWT and WTC analyses, using a widely used standardization procedure (zero mean, unit standard deviation) (e.g. Torrence and Compo, 1998; Grinsted et al., 2004; Labat, 2005)."

Please also note the supplement to this comment:
https://www.hydrol-earth-syst-sci-discuss.net/hess-2018-491/hess-2018-491-AC1-supplement.pdf

───────────────────────────

---

## Author Comment (AC2) · 1 Mar 2019

This study evaluates the inïˇnĆuence of low-frequency oscillations linked to large-scale oceanographic-atmospheric processes, on streamïˇnĆow variability in small tropical coastal mountain rivers of the Sierra Nevada de Santa Marta, Colombia. By using spectral analysis and Hilbert Huang transform, the study aims to (1) explore temporal characteristics of streamïˇnĆow variability, (2) estimate the net contribution to the

energy spectrum of low-frequency oscillations to streamïˇnĆow anomalies, and (3) analyze the linkages between streamïˇnĆow anomalies and large-scale, low-frequency oceanographic/atmospheric processes.

The main topic of the article is important to Hydrology and water resource management, and deserves to be published in HESS. However, the results need to be discussed in a broader context, comparing the main ïˇnĄndings with related literature. The tools applied to address the research questions are adequate and properly applied, however some technical details are necessary to be described. In addition, a deeper explanation about the physical mechanisms linking PDO, AMO, TNA and the basins' hydrology is necessary. Also, the whole subject is about the possibility of a cause-effect relation between decadal oscillations and streamïˇnĆow, but the concept of phase locked signals is completely missing in the interpretation of the results and the discussion, which I think is necessary. Thus, my decision is accepted with major revisions.

Authors: We appreciate your comments and the overall assessment of the work. General adjustments will be made in the manuscript and especially in the discussion section (see details below - Author's response (AC) and Author's Changes in Manuscript (ACM)):

(AMC) In order to highlight the effect of superimposed signals on streamflow variability, the differences in the phase relationship when comparing high and low frequencies, the phase relationships between streamflow and climatic/oceanographic índices, the manuscript will be modified as follows (new text in bold and cursive):

Discussion (Page 11 – Line 30): "The maximum intensity of the inter-annual signal, which occurred between 1998 and 2002 in most rivers, also coincides with the interval of greater intensity of the quasi-decadal signal (1998-2005) (Fig. 2). Streamflow rates also exhibit inflection points in their trends between the 1990s and 2000s, a period that also coincides with the increase in the amplitude of low-frequency oscillations

(Fig. 4). These results show that the superposition of climatic / oceanographic signals, particularly the modulation of the effects of the interannual signal due to phase changes in long period signals, is a key element within the occurrence of extreme events at sub-regional scale (i.e. Steinman et al., 2014; Shi et al., 2017; Murgulet et al., 2017; Su et al., 2018)."

Discussion (Page 12 – Line 13): "These results suggest a relation between changes of these climatic/oceanographic indexes and long-term streamflow variability, indicating that these watersheds are sensitive to changes in the background climate state. Furthermore, power and phase relationships between streamflow and different indices (Fig. 6-8) were relatively steady for low-frequencies (i.e. > 96 months) but unstable and disperse for high-frequencies (i.e. < 96 months). Such differences in these patterns, suggest that during longer periods, streamflow might be modulated by the slowly change in the climate background state; whereas during shorter periods, the streamïñĆow is not only controlled by large-scale ocean–atmosphere patterns, but also by local short-term phenomena. This result highlights, once again, the significant effect of the superposition of signals of different frequencies in the streamflow variability (e.g. Pasquini and Depetris, 2007; Labat, 2010; Steinman et al., 2014; Shi et al., 2016; Murgulet et al., 2017). For lower frequencies, in both the XWT and WTC analysis, the phase relationship exhibited a stable phase lag inside the significance common power regions for each river (Fig. 6-8). Such consistent or slowly varying phase lag, imply a phase-locked relationship and establish a physically link (i.e. not a casual relationship) between the streamflow variability and each climatic/oceanographic indices (Grinsted et al., 2004; Labat, 2005). Outside the areas with signiïñĄcant power the phase relationship changed (Fig. 6-8). We therefore speculate that despite the relatively strong link between streamflow and these indices at specific frequencies (low) and temporal windows (Fig. 6-8), these relationships are highly non-linear and non-stationary; depending heavily on the phase experienced by these oscillations and their dynamic feedback processes (e.g. Battisti and Sarachick, 1995; Einfeld and Alfaro, 1999; Garreaud et al., 2009). Differences in spectral correlations between rivers from

the western and the eastern slopes, and differences in phase relationships observed in some rivers, indicate that further research is required to draw conclusions about the specific drivers of low-frequency variability.

Discussion (Page 12 – Line 13): "Although robust hypotheses have been put forth regarding the physical relation between the PDO (Poveda, 2004), the AMO (Arias et al., 2015) and the TNA (Enfield and Alfaro, 1999) and the climate of northwestern South America, the physical mechanisms by which these phenomena influence the hydrology at low-frequency scales remains elusive. Understand the specific physical links between streamflow varibility and these climatic/oceanographic indices is beyond the aim of this study. Nevertheless, we believe these mechanisms may relate to SST gradients between the Pacific and Atlantic oceans".

SpeciïñĄc comments: (1) First paragraph: a more in-depth description on the PDO-ENSO relation is necessary in addition to AMO and TNA relations to inter-annual oscillations.

(AR) Adjustments will be made in the first paragraph to highlight the telleconections and interactions that exist between these phenomena.

(ACM) In order to provide a more in-depth description of the phenomenon interactions, the manuscript will be modified (Page 2 – Line 2) as follows (new text in bold and cursive):

First paragraph (Page 2 – Line 2): "In the past several decades, streamflow variability has increased (Milliman et al., 2008; Dai et al., 2009), causing frequent and pronounced flood/drought cycles (Hungtinton, 2006). Atmospheric and oceanographic processes are major sources of streamflow variability (Jhonson et al., 2013; Schulte et al., 2016). The El Niño-Southern Oscillation (ENSO) is among the most prevalent oceanographic/atmospheric processes linked to streamflow variability in tropical and subtropical areas (Battisti and Sarachick, 1995; Amarasekera et al., 1997; García and Mechoso, 2005, Labat, 2010). ENSO, however, is also affected by longer-period

changes in the background state (Garreaud et al., 2009; Chowdary et al., 2014). It has been pointed out that its effects can be modulated by the coupling that exists between ENSO phases and long period events, such as the Pacific Decadal Oscillation (PDO) and the Atlantic Meridional Oscillation (AMO) (i.e. Brown and Comrie, 2004; Murgulet et al., 2017; Shi et al., 2017). For example, the 1997-1998 El Niño event occurred during a PDO shift from a warm to a cold phase, but recent warming (2010-2011) in the Pacific occurred during a cold phase of the PDO. Multiple atmospheric / oceanograhic oscillations collectively impose a more complex influence on hydrology (Labat, 2010; Nalley et al., 2016; Shi et al., 2016). Thus, changes in the intensity and frequency of extremes events depend on the coupling and teleconnection of these large-scale atmospheric/oceanographic processes. Overall, such interactions occur through changes in the sea level pressure (SLP) and sea surface temperature (SST) gradients, which in turn lead to flux changes in the atmosphere (ie Einfeld and Alfaro, 1999, Jhonson et al., 2013, Sagarike et al., 2015, Murgulet et al., 2017, Shi et al, 2017). Such atmospheric and oceanographic interactions, as well as their role in hydrological variability, have gained attention in recent years (Tootle et al., 2008; Arias et al., 2015; Sagarika et al., 2015; Nalley et al., 2016). Thus, a major question in the study of hydrology is the potential effect of longer-period climate modes on the strength of a particular El Niño/Niña event. The interplay that exists between the multiple large-scale oscillations and the regional hydrological process constitutes a complex climate-land coupled system (Steinman et al., 2014; Murgulet et al., 2017)".

(2) - Second paragraph: the main idea is confusing. Maybe split paragraphs one for novel statistical methods and another related to the hydrology in Colombia.

(AR) Adjustments will be made on the manuscript.

(ACM) In order to avoid confusión in the ideas expressed in the second paragraph, the paragraph will be split as follows (new text in bold and cursive):

Second paragraph (Page 2 – Line 16): "Several authors have examined the relationship between streamflow variability in northern South America and large-scale oceanographic/climate indices, particularly those linked to ENSO (e.g. the Southern Oscillation Index [SOI], the Multivariate ENSO Index [MEI], and Niño 1, 2, 3, 4) (Robertson and Mechoso, 1998; Hastenrath, S., 1990; Gutiérrez and Dracup, 2001; Poveda et al., 2001; Restrepo and Kjerfve, 2004; García and Mechoso, 2005). New variables such as SST gradients in the Caribbean Sea and low-frequency oscillations, together with new statistical methods (e.g. Singular Value Decomposition and Principal Components Analyses) are now used in streamflow analysis. These new approaches have improved hydrological forecast models, compared to predictions based solely on El Niño-based indices. For example, such an approach allowed to establish that the extremely anomalous wet seasons in northern South America between 2010 and 2012 were not only associated with ENSO anomalies, but also with an enhanced Atlantic Meridional Mode (AMO), a low-frequency oscillation that is independent of ENSO (Arias et al., 2015). The new models also reduce the spatial bias of SST, which affects hydrology at regional scales (Tootle et al., 2008; Córdoba-Machado et al., 2016). These studies, however, failed to include representative small basins (area $\leq$ 5000 km2) that drain into the Caribbean Sea in northern South America. Furthermore, mountain rivers flowing from the Sierra Nevada de Santa Marta (SNSM) massif (Fig. 1, Table 1) are absent from these models. Pierini et al. (2015) indicated that rivers from the SNSM exhibit a distinctive hydrological pattern, which differs from that of other rivers in northwestern South America. Differences are especially pronounced between rivers in the SNSM and those with headwaters in the Colombian Andes. The main difference lies in the relatively low contribution from ENSO-related oscillations to the net streamflow variability exhibited by SNSM rivers (Restrepo et al., 2012, 2014). Overall, contribution from low-frequency oscillations to streamflow variability is poorly understood, particularly in small, tropical, coastal mountain rivers (Stevens and Ruscher, 2014; Nalley et al., 2016; Marini et al., 2016). These fluvial systems possess low streamflow buffering capacity because of their topographic setting (Milliman and Syvitski, 1992), and they are exposed to regional-scale atmospheric/oceanographic processes (Hastenrath, 1990; Enfield and

Alfaro, 1999). Furthermore, it has been established that changes in the Caribbean SST gradients affect the amount of rainfall in northern South America (Enfield and Alfaro, 1999), but there is no evidence that such changes affect the hydrological variability of SNSM rivers, which are characterized by a limited ability to filter hydrological signals (Restrepo et al., 2014)".

Third paragraph (Page 3 – Line 8): "Standard statistical techniques are generally unable to explain the complex interactions, based on non-linear and non-stationary underlying processes, among a wide range of climatic/oceangrapic oscillations and their associated effects on hydrolgy (i.e. Grinsted et al., 2004; Xu et al., 2004; Labat, 2005; Shi et al., 2017). Spectral analyses such as Wavelet Transform (WT) and the Hilbert Huang Transform (HHT) (Grinsted et al., 2004; Labat et al., 2005; Torrence and Compo, 2008; Massei and Fournier, 2012; Schulte et al., 2016) have proven useful to identify the timing of important features of non-stationary signals and to discriminate the relative contribution of signal components, which may change through time. The objectives of this study were to: (1) explore the temporal characteristics of streamflow variability, with emphasis on low-frequency oscillations, (2) estimate the net contribution (i.e. energy spectrum) of such oscillations to streamflow anomalies, and (3) analyze the linkages between streamflow anomalies and large-scale, low-frequency, oceanographic/atmospheric processes (Table 2) in small, tropical, coastal mountain rivers of the SNSM (Fig. 1 and Table 1). To our knowledge, this is the first study to estimate the contribution of low-frequency oscillations to the hydrologic variability at a subregional scale and in these type of watersheds (i.e. small, coastal, and mountainous), and specifically in northern South America, where ENSO has been identified previously as the preeminent driver on streamflow variability (i.e. Gutierrez and Dracup, 2001; Poveda et al., 2001; Córdoba-Machado et al., 2016)".

(3) - Third paragraph: to keep the logic of the manuscript the main objectives ought to be aligned with the sub-sections presented in section 4.

(AR) Yes, it is true. The manuscript will be adjusted following this suggestion.

(ACM) In order to align the main objectives and subsequent sub-sections, the manuscript will be modified (Page 3 – Line 8) as follows:

Text to be removed is highlighted in bold and cursive: "The objectives of this study were to: (1) study the influence of low-frequency oscillations (linked to large-scale oceanographic/atmospheric processes) on streamflow variability, (1) explore the temporal characteristics of streamflow variability, (2) estimate the net contribution (i.e. energy spectrum) of low-frequency oscillations to streamflow anomalies, and (3) analyze the linkages between streamflow anomalies and large-scale, low-frequency, oceanographic/atmospheric processes (Table 2) in small, tropical, coastal mountain rivers of the SNSM (Fig. 1 and Table 1)".

New text in the manuscript (new text in bold and cursive): "The objectives of this study were to: (1) explore the temporal characteristics of streamflow variability, with emphasis on low-frequency oscillations, (2) estimate the net contribution (i.e. energy spectrum) of such oscillations to streamflow anomalies, and (3) analyze the linkages between streamflow anomalies and large-scale, low-frequency, oceanographic/atmospheric processes (Table 2) in small, tropical, coastal mountain rivers of the SNSM (Fig. 1 and Table 1)".

(4) - Pag. 4 line 30: explain the main difference between XWT and WTC.

(AR) On Page 5 - Line 13 we present a brief description of the XWT and WTC highlighting their differences. However, based on your observations, we consider it relevant to highlight the difference between these methods on section 3.1

(ACM) In order to reinforce the main difference between XWT and WTC, the manuscript will be modified (Page 4 – Line 29) as follows (new text in bold and cursive):

"We also used Cross Wavelet Transform (XWT) and Wavelet Coherence (WTC) to estimate the correlation between streamflow and eight large-scale climate/oceanographic processes (Table 2). The XWT unveils high common powers and relative phases in a

time-frequency space; whereas the WTC finds significant coherence even with a low common power, and shows confidence levels against red noise, highlighting locally phase locked behaviors (Shumway and Stoffer, 2004; Grinsted et al., 2004; Labat, 2005).

(5) - Pag. 7 line 10: equation (7) may be wrong.

(AR) Yes, there was a imprecision in equation (7). There is a term in the denominator that must be removed.

(ACM) The manuscript will be modified in order to adjust equation (7) (Page 7 – Line 10) as follows (new text in bold and cursive):

f ÌĚ(n)=(∫ _Ⓗ∞▒ãĂŰãĂŰfEãĂŮ_n (f)dfãĂŮ)/(∫ _Ⓗ∞▒ãĂŰE_n (f)dfãĂŮ) (see attached file) (7)

(6) - Pag. 8 line 6-7: from Fig. 2 the statement is not evident for station Frío, please explain.

(AR) This comment is very pertinent. When referring to the simultaneously ocurrence of different signal bands, the Frío River exhibits a jointly oscillation of the annual and quasi-decadal bands during 1988 and 1990, and of the annual, interanual and quasi-decadal bands between 19988 and 2002. However, as you stated, the interaction of the cuasi-biennal, annual, interanual and cuasi-decadal bands between 2008 and 2012 is not completely evident, because the cuasi-biennal bands exhibit moderate power (mild yellow) instead of high power (intense yellow) during this period. Also the streamflow record of the Frío River ends in 2009. Thus, is not appropriate to make inferences beyond this date.

(ACM) In order to clarify the examples on the simultaneously ocurrence of different signal bands in the Frío River, the manuscript will be modified (Page 8 – Line 6) as follows:

Text to be removed is highlighted in bold and cursive: "A quasi-biennial oscillation occurred jointly with annual, inter-annual and quasi-decadal oscillations during the 2008-2012 interval, in the Fundación, Aracataca, Frío and Palomino Rivers (Fig. 2)".

New text in the manuscript: "A quasi-biennial oscillation occurred jointly with annual, inter-annual and quasi-decadal oscillations during the 2008-2012 interval, in the Fundación, Aracataca and Palomino Rivers (Fig. 2)".

Please also note the supplement to this comment:
https://www.hydrol-earth-syst-sci-discuss.net/hess-2018-491/hess-2018-491-AC2-supplement.pdf

---

## Author Response (AR1)

In general, reviewers requested to (1) discuss the scientific novelty of this study and (2) have a depeer discussion on the phase relationships between the flow variability and the low frequency oceanographic/climatic oscillations (i.e. AMO, PDO and TNA). All questions and comments from reviewers were answered and uploaded to the system on March 1st 2019.

We now attach the revised manuscript, which not only incorporates answers to comments and recommendations raised by the reviewers, but also includes answers to comments presented by the Editor in his most recent communication. With respect to the editor comments we would like to highlight here some answers to his comments:

**Hypothesis and objectives**

we think this paper deals with a relevant topic in the field of surface hydrology as the reviewers pointed out. The analysis of the effect of large-scale atmospheric/oceanographic oscillations (particularly those of low frequency) on local hydrology variability and on the modulation of other drivers that have received the most attention, such as the ENSO, allows to address questions of scientific relevance.

*Page 2 - Line 11: Multiple atmospheric/oceanographic oscillations collectively impose a more complex influence on hydrology (Labat, 2010; Nalley et al., 2016; Shi et al. , 2017). Thus, changes in the intensity and frequency of extreme events depend on the coupling and teleconnections of these large-scale atmospheric/oceanographic processes.*

*Page 2 - Line 18: ... a major question in the study of hydrology is the potential effect of longer-period climate modes on the variability of hydrological processes and on the strength of a particular event, such as El Niño / Niña. That exists between the multiple large-scale oscillations and the regional hydrological processes at a complex climate-land coupled system (Steinman et al., 2014, Murgulet et al., 2017)*

There is not much information about the effect of low frequency large-scale atmospheric/oceanographic oscillations on the hydrological variability at the sub-regional scale (especially in Northern South America), where it has been pointed out that the ENSO is the pre-eminent element of such variability.

**Page 2 – Line 3**: *The El Niño-Southern Oscillation (ENSO) is among the most prevalent oceanographic/atmospheric processes linked to streamflow variability in tropical and subtropical areas (Battisti and Sarachick, 1995; Amarasekera et al., 1997; García and Mechoso, 2005, Labat, 2010)*

**Page 2 – Line 22**: *Several authors have examined the relationship between streamflow variability in northern South America and large-scale oceanographic/climate indices, particularly those linked to ENSO (e.g. the Southern Oscillation Index [SOI], the Multivariate ENSO Index [MEI], and Niño 1, 2, 3, 4) (Robertson and Mechoso, 1998; Hastenrath, S., 1990; Gutiérrez and Dracup, 2001; Poveda et al., 2001; Restrepo and Kjerfve, 2004; García and Mechoso, 2005)*

*Page 3 – Line 7: Overall, contribution from low-frequency oscillations to streamflow variability is poorly understood, particularly in small, tropical, coastal mountain rivers (Stevens and Ruscher, 2014; Nalley et al., 2016; Marini et al., 2016).*

Due to the differences that have been established previously between the rivers of the Sierra Nevada de Santa Marta (SNSM) and other rivers that drain the northwest of South America, our work suggests that there is a discernible and quantifiable effect of the low frequency oscillations in the patterns of streamflow variability of the SNSM rivers.

*Page 3 – Line 1: Pierini et al. (2015) indicated that rivers from the SNSM exhibit a distinctive hydrological pattern, which differs from that of other rivers in northwestern South America. Differences are especially pronounced between rivers in the SNSM and those with headwaters in the Colombian Andes. SNSM rivers exhibits a relatively low contribution from ENSO-related oscillations and a larger influence of quasi-decadal oscillations in their streamflow variability signals, compared to Andean rivers (Restrepo et al., 2012, 2014)*

*Page 3 – Line 6: The magnitude of such influence and its link with large-scale climatic/oceanographic oscillations is still unknown*

The objectives of this study are aimed at verifying this approach (i.e. testable hypothesis). Each of the sub-sections of the results section is aimed at developing these objectives.

*Page 3 – Line 21: (1) explore the temporal characteristics of streamflow variability, with emphasis on low-frequency oscillations, (2) estimate the net contribution (i.e. energy spectrum) of low-frequency oscillations to streamflow anomalies, and (3) analyze the linkages between streamflow anomalies and large-scale, low-frequency, oceanographic/atmospheric processes (Table 2) in small, tropical, coastal mountain rivers of the SNSM (Fig. 1 and Table 1)*

**Experimental design and data analysis techniques.**

Methods used in this work (i.e. Spectral Wavelet Analysis and Hilber Huang Transform) were selected based on their ability to process non-stationary data that respond to multiple and diverse factors (i.e. streamflow time series). Appropriate methods are considered to develop the objectives outlined in the Introduction

*Page 5 – Line 17: We used Continuous Wavelet Transform (CWT) and Hilber Huang Transform (HHT) analyses to estimate periodicities, variability patterns (**Objective 1**), and the net contribution (i.e. energy spectrum) of low-frequency oscillations to streamflow anomalies (**Objective 2**). We also used Wavelet Coherence (WTC) and Cross Wavelet Transform (XWT) to estimate the correlation between streamflow and eight large-scale climate/oceanographic processes (**Objective 3**).*

These methods offer great advantages over conventional statistical methods and have therefore been widely used for processing and analysis of geophysical data (eg Long et al.,

1995, Huang et al., 1999, Grinsted et al., 2004; Labat et al., 2005; Pasquini and Depetris, 2007; Torrence and Compo, 2008; Labat, 2010; Barnhart, 2011; Massei and Fournier, 2012; Jhonson et al., 2013; Nulley et al., 2016; Schulte et al. ., 2016). The Continuous Wavelet Transform (CWT) and Hilbert Huang Transform (HHT) analyzes are robust, sufficient and widely tested methods to address objectives such as those proposed in this paper, as the evaluators pointed out in their comments.

***Page 3 – Line 15****: Standard statistical techniques are generally unable to explain the complex interactions, based on non-linear and non-stationary underlying processes, among a wide range of climatic/oceanographic oscillations and their associated effects on hydrology (i.e. Grinsted et al., 2004; Xu et al., 2004; Labat, 2005; Shi et al., 2017). Spectral analyses such as Wavelet Transform (WT) and the Hilbert Huang Transform (HHT) (Grinsted et al., 2004; Labat et al., 2005; Torrence and Compo, 2008; Massei and Fournier, 2012; Schulte et al., 2016) have proven useful to identify the timing of important features of non-stationary signals and to discriminate the relative contribution of signal components, which may change through time*

The approach and experimental design of this study is not novel, since the paper applies established methods and it follows ideas that other papers have developed/applied (e.g. Grinsted et al., 2004; Labat et al., 2005; Pasquini and Depetris, 2007, Torrence and Compo, 2008, Massei and Fournier, 2012, Restrepo et al., 2014, Schulte et al., 2016, Valdes-Pineda et al., 2017). This aspect is pointed out by reviewers. The Data and Methods section provides information on the robustness of these analyzes, their ability to test hypotheses in geophysical data and the statistical precision estimators used so that the patterns and relationships identified are not the result of chance and/or randomness, but of physical links.

***Page 6 – Line 19****: A value of 6 was defined for the frequency localization of the Morlet wavelet (ωo) to fulfill the admissibility condition (localization in time and frequency, zero mean, and to acquire a proper balance between frequency and time) (Torrence and Compo, 1998; Grinsted et al., 2004; Nalley et al., 2016). The 95% confidence level was calculated for contours and edge effects area after the method of Torrence and Compo (1998). The edge effect was addressed by the zero-padding approach. This procedure creates discontinuities at both ends of the data, particularly at larger scales. The power displayed in this area is expected to be weaker than actually shown (Nalley et al., 2016). The area in the WT spectrum where the edge effect is shown is referred to as the Cone of Influence (COI). The interpretation of the WT power spectra was limited to the area outside the COI, thus the COI is represented by the region outside of the concave-up area*

***Page 6 – Line 11****: An in-phase relationship is indicated by arrows in the enclosed significant regions of the XWT and WTC spectra that point straight to the right. On the other hand, and anti-phase relation is indicated by arrows pointing straight to the left. Arrows that do not point straight to the right or left indicate a lead/lag relationship, when a climate/oceanographic index led the streamflow response (Grinsted et al., 2004; Nalley et al., 2016)*

***Page 9 – Line 12****: Thus, it is likely that longer time series are required to test the low-frequency oscillations statistical significance within the global wavelet spectrum. Information on these low-frequency oscillations was considered useful because (1) the zero-padding technique reduces the lower frequencies true power, (2) the CWT isolates hidden signals*

*not shown by other techniques, and (3) they are within the range defined by edge effects and cut-off frequency*

***Page 9 – Line 21****: Information on the last IMF mode of Fundación (C7) and Gaira (C7) Rivers must be analyzed cautioulsy as they are outside the range established for the edge effects approach (Table 4)*

***Page 13 – Line 22****: Such consistent varying phase lag, implies a phase-locked relationship and suggests a physically link (i.e. not a casual relationship) between the streamflow variability and each of the climatic/oceanographic indices (Grinsted et al., 2004; Labat, 2005)*

**Discussion**

The novelty of this study lies, fundamentally, in the location (Caribbean) and basins' physiography from which the analyzed data come from (**Page 2- Line 31, Page 3 - Line 1**).

These unique charactersitics allowed us to:

(1) highlight the influence of low frequency climate indices (ie PDO, AMO and TNA) on the surface hydrology of northern South America (where its effect had previously been minimized - **Page 13 - Line 1**)

(2) provide strong evidence that low frequency oscillations are major players on the streamflow variability for this area. Its magnitude is of the same order (or higher in some cases) than that of the ENSO (considered as the main driver of the superficial hydrology of Northwestern Southamerica) (**Page 11 - Line 3, Page 11 - Line 7**)

(3) show that flow variability is a consequence of the concurrence of different frequency signals, rather than to a specific signal. This also highlights the modulating effect of quasi-decadal signals (**Page 11 - Line 16**)

(4) understand the role of low frequency oscillations in the streamflow of basins with a small drainage area. These signals had shown less intensity in regional scale studies. (i.e. Murgulet et al., 2017) (**Page 11 - Line 11**).

All these aspects, together with the limitations of this work, were highlighted in the discussion and conclusions sections.